# The Correlated Beta Dose Optimisation Approach: Optimal Vaccine Dosing Using Mathematical Modelling and Adaptive Trial Design

**DOI:** 10.3390/vaccines10111838

**Published:** 2022-10-30

**Authors:** John Benest, Sophie Rhodes, Thomas G. Evans, Richard G. White

**Affiliations:** 1Department of Infectious Disease Epidemiology, London School of Hygiene and Tropical Medicine, Keppel Street, London WC1E 7HT, UK; 2Vaccitech Ltd., The Schrodinger Building, Heatley Road, The Oxford Science Park, Oxford OX4 4GE, UK

**Keywords:** adaptive design, clinical trials, continual modelling, dose response, dosing, modelling, non-parametric models

## Abstract

Mathematical modelling methods and adaptive trial design are likely to be effective for optimising vaccine dose but are not yet commonly used. This may be due to uncertainty with regard to the correct choice of parametric model for dose-efficacy or dose-toxicity. Non-parametric models have previously been suggested to be potentially useful in this situation. We propose a novel approach for locating optimal vaccine dose based on the non-parametric Continuous Correlated Beta Process model and adaptive trial design. We call this the ‘Correlated Beta’ or ‘CoBe’ dose optimisation approach. We evaluated the CoBe dose optimisation approach compared to other vaccine dose optimisation approaches using a simulation study. Despite using simpler assumptions than other modelling-based methods, we found that the CoBe dose optimisation approach was able to effectively locate the maximum efficacy dose for both single and prime/boost administration vaccines. The CoBe dose optimisation approach was also effective in finding a dose that maximises vaccine efficacy and minimises vaccine-related toxicity. Further, we found that these modelling methods can benefit from the inclusion of expert knowledge, which has been difficult for previous parametric modelling methods. This work further shows that using mathematical modelling and adaptive trial design is likely to be beneficial to locating optimal vaccine dose, ensuring maximum vaccine benefit and disease burden reduction, ultimately saving lives

## 1. Introduction

Vaccines are an effective tool in global disease burden reduction. The amount of vaccine given to an individual (the ‘dose’) is a key decision in vaccine development to ensure an effective vaccine campaign. Dose can affect the efficacy, toxicity and cost associated with vaccine rollout [1,2,3]. However, selecting optimal dose (‘dose optimisation’) is non-trivial [4,5,6]. Vaccine dose-ranging trials are typically small (<100 individuals) [7,8,9,10], limiting the amount of data that can be used for dose decision making. In addition, vaccine dose-ranging clinical trials need to be conducted such that not only are useful data gathered, but also such that the interests and safety of the trial participants are respected [11].

In order to select optimal vaccine dose within the constraints of small trial sizes and ethical trial design, mathematical modelling and adaptive clinical trial design have been suggested. Previous work into mathematical modelling has shown promise for accelerating and improving dose decision making in vaccine development [2,12,13]. Whilst making dosing decisions based on modelling is common in drug development, these methodologies are not yet utilised to the same extent within vaccine development [12,14,15]. Further, adaptive trial design has also been suggested to be effective for the purpose of selecting optimal doses [16,17,18]. Here, modelling or statistical analysis is conducted at interim time points to maximise the proportion of trial participants that receive near optimal doses. Adaptive design may lead to more optimal dose selection and more ethical clinical trials.

Vaccine dose–response mathematical models are systems of equations that are used to describe the relationship between vaccine dose and vaccine response. This requires making assumptions regarding which models can accurately describe vaccine dose–response. Previous work has shown that for some vaccines an increase in dose leads to increased efficacy responses, but that for other vaccines there is a maximum efficacy dose after which an increased dose leads to decreased vaccine efficacy [13,19,20].This means that there may be uncertainty in the correct models to use. Selecting optimal vaccine dose using models which are ‘misspecified’, meaning they are not appropriate for describing the dose–response relationship for the purposes of selecting optimal dose, could lead to suboptimal vaccine dosing, decreasing efficacy or increasing toxicity [21,22,23].

We have previously discussed the use of model averaging to account for this uncertainty [21]. Alternatively, others have suggested that non-parametric models can be effective for locating optimal dose in the case of model uncertainty [24,25,26].Whilst the assumption of parametric models is that vaccine dose–response follows some pre-specified equation/shape, non-parametric models do not assume a predefined equation/shape.

One type of non-parametric model is the Continuous Correlated Beta Process (CCBP) model [27]. This is a form of non-parametric mathematical model that has previously been discussed for automated stroke rehabilitation and modelling of genetic ancestry [28,29]. CCBP models have the properties of being simple to implement, interpret and update based on available data, and do not require the assumption of a specific dose–response shape. The modelling assumption is instead that “similar” doses will cause “similar” responses. We hypothesised that the application of CCBP models in an adaptive trial design setting may be an effective approach for conducting clinical trials to select optimal vaccine dose. We call this Correlated Beta (CoBe) dose optimisation.

We evaluated this novel dose-optimisation approach in potential application to four potential open topics in mathematical modelling for optimal vaccine dose selection. Firstly, selection of a maximally efficacious vaccine dose given uncertainty in dose-efficacy curve shape. Secondly, how to locate the maximally efficacious doses for prime-boost paradigm vaccines. Thirdly, optimal vaccine dose selection that includes multiple objectives, such as both maximising efficacy and minimising toxicity. Fourthly, how can expert knowledge be incorporated into vaccine dose modelling. We hoped that this novel modelling approach could have potential for practical application over a number of vaccine use cases, and that the highlighted model could provide interpretable quantitative insight for vaccine developers.

In this work we aimed to use simulation of dose-finding clinical trials to assess the use of the ‘Correlated Beta dose optimisation approach’ in selecting optimal vaccine dose. To answer the questions posed above, we investigated the CoBe dose optimisation approach relative to three other dose optimisation approaches (DOAs).

A ‘Parametric’ DOA that used parametric modelling and adaptive trial designAn ‘Adaptive Naive’ DOA that used adaptive trial design but not modelling.A ‘Uniform Naive’ DOA that used neither adaptive trial design nor modelling.

To perform this analysis, we simulated a large number of clinical trials for a large number of qualitatively different ‘scenarios’, each representing different ‘true’ dose-efficacy or dose-efficacy and dose-toxicity relationships. We considered not only the quality of the final selected dose but also the benefit to clinical trial participants for all four DOAs for clinical trials with between 6–300 total trial participants.

Specifically, to address to above questions, our objectives were to:Evaluate the Correlated Beta Dose Optimisation Approach for optimising vaccine efficacy for a single dose administration.Evaluate the Correlated Beta Dose Optimisation Approach for optimising vaccine efficacy for a prime-dose/boost-dose administration.Evaluate the Correlated Beta Dose Optimisation Approach for optimising vaccine utility, maximising efficacy, and minimising toxicity.We also include a fourth objective which considered only the CoBe DOAEvaluate the use of expert knowledge informed Continuous Correlated Beta Process priors for vaccine dose-optimisation.

## 2. Materials and Methods

In very high-level summary, we used a simulation study methodology [30,31,32,33] to evaluate the novel Correlated Beta (CoBe) dose optimisation approach with regard to several open topics in vaccine dose optimisation and provide a comparative evaluation relative to other potential dose-optimisation approaches that could be used to select optimal vaccine dose. This work is summarised in Figure 1.

This methods section is split into four sections. In Section 1, we defined the concept of ‘optimal vaccine dose’ and of ‘dose optimisation approaches’ (DOAs). In Section 2, we defined and described the Correlated Beta (CoBe) DOA that was the focus of this work, along with the three other DOAs that were investigated in this work in comparison to the CoBe DOA. In Section 3, we describe the simulation study methodology that was used to evaluate and compare these DOAs. Section 3 also contains description of the metrics used to evaluate the DOAs with regard to their potential effectiveness for optimization of vaccine dose and benefit to trial participants, and details of the simulation study that would be required to replicate this work. Finally, in Section 4 we describe how we investigated the four objectives of this work using the concepts and terminology developed in the previous three sections.

### 2.1. Section 1. Definition of the Concepts of ‘Optimal Vaccine Dose’ and ‘Dose-Optimisation Approaches’

#### 2.1.1. Definition of ‘Optimal Vaccine Dose’

In this work optimal dose was defined as the dose that maximises some utility function $\;U\left(p_{eff},p_{tox}\right)$ where $p_{eff}$ and $p_{tox}$ are binary efficacy and toxicity probabilities that are dependent on vaccine dose.

Throughout this work we will consider only two utility functions, one that aims to maximise efficacy (‘Maximum Efficacy’) versus dose and literature informed utility function that balances maximising efficacy and minimising toxicity (‘Utility Contour’, as used in [34,35]) versus dose. Formally these are

Maximum Efficacy:(1)$$U\left(p_{eff},p_{tox}\right)=p_{eff},$$

Utility Contour:(2)$$U\left(p_{eff},p_{tox}\right)=1-{\left({\left(\frac{1-p_{eff}}{1-anchor_{eff}}\right)}^{rho}-{\left(\frac{p_{tox}}{anchor_{tox}}\right)}^{rho}\right)}^{\frac{1}{rho}},$$
where $anchor_{eff}$, $anchor_{tox}$ and rho are parameters defined by clinicians to weight the relative importance of efficacy to toxicity (see S1 and [35] for more detail). Optimal dose was constricted to the dosing domain.

##### Dosing Domain

The possible doses that can be selected for testing or predicted as optimal was called the ‘dosing domain’. Dosing domains are generally continuous in nature, though are often discretized to a finite number of possible doses for the purpose of optimisation and due to potential practical limitations [36]. We will only consider discretized dosing domains in this work.

Previous work has investigated mathematical modelling for the selection of optimal dose with regard to a single-administration vaccine [2,13,21]. In this work we would also like to consider optimising dose ‘prime/boost’ paradigm vaccines, which are vaccines that are administered as two or more doses at separate time points [37,38]. Here, doses in the dosing domain are possible combinations of possible doses for each prime or boost administration.

#### 2.1.2. Definition of a ‘Dose-Optimisation Approach’

A dose-optimisation approach (DOA) is the combination of methods used to design clinical trials/choose the doses that trial participants will receive, along with the methods used to select ‘optimal’ dose based on the resulting data. We focus here on ‘continual modelling’ DOAs, where modelling is conducted at interim stages of the trial and used to guide selection of the next trial doses.

For this work, a DOA consists of

A model for vaccine dose-efficacy and/or dose-toxicity.A method of trial dose selection: How doses are chosen during the trial.A method of final dose selection: How to choose the dose that would be continued forward to further research or clinical use.A choice of how to discretize the dosing domain: Whether there was a small or large number of doses that could be tested, further detail in 2.3.3.1. This was previously discussed by [36,39].

### 2.2. Section 2. Definition of the Correlated Beta (CoBe) Dose-Optimsation Approach and Three Other Dose-Optimisation Approaches That Were Investigated in This Work

#### 2.2.1. Model for Vaccine Dose-Efficacy and/or Toxicity: Continuous Correlated Beta Processes

The CoBe DOA uses Continuous Correlated Beta Process (CCBP) models [27] to model vaccine dose-efficacy/toxicity. These are not only simple to implement but can be extended to prime/boost dose–response problems (Objective 2) or extended to include expert prior predictions (Objective 4). In this section we discuss the intuition and implementation of Continuous Correlated Beta Processes (CCBP).

In contrast to parametric models, which assume some curve shape can describe vaccine dose-efficacy or dose-toxicity, the CCBP models defined here do not assume a specific shape, and instead make a simpler assumption; ‘similar doses yield similar responses’. CCBP have been described previously in detail in the context of modelling for automatic stroke rehabilitation [27,28,29]. The two main elements of a CCBP model are Beta distributions and correlation kernel functions.

##### Beta Distributions

Beta distributions describe a probability distribution of probabilities for binary outcomes [40]. Suppose that we would like to know the probability of some response (efficacy or toxicity in this work) being observed for a vaccine administered at a pre-chosen dose. We call this probability $\;p_{response}$, and we have no prior expectation for what the true value of $p_{response}$ is. Suppose that after a trial of 1 individual we have observed 1 responder (and hence 0 non-responders). The maximum likelihood estimate of $p_{response}$ given by these data would be $p_{response}=1.0$. However, $p_{response}=0.9\;$ would also intuitively be a reasonable guess and $p_{response}=0.1$ would be much less probable (Figure 2). Beta distributions allow for a formalised description of the probability of a certain probability of response given the observed data.

A beta distribution is defined by two parameters, α and β. We write
(3)$$p_{i,r}^{n}\sim Beta\left(\alpha_{i,r}^{n},\beta_{i,r}^{n}\right),$$
to say that the probability of observing response r for some dose $d_{i}$ based on the first n data points can can be described by a beta distribution with parameters $\alpha_{i,r}^{n}$ and $\beta_{i,r}^{n}$. Increasing $\alpha_{i,r}^{n}$ shifts the beta distribution towards higher $p_{i,r}^{n}\;$ and increasing $\beta_{i,r}^{n}$ shifts the beta distribution towards lower $p_{i,r}^{n}$. Increasing either of these parameters reduces the confidence intervals of the probability distribution. See Figure 3 for a visualisation of this. In this work, response r can be efficacy or toxicity.

##### Updating Beta Distributions

As we aim to run multiple trials over time, we can use the data gathered to update our beta distributions to give us a better idea of optimal dose. Algorithm 1 shows the update rule for updating the α and β parameters of beta distributions after observing data.

Note that this update rule means our understanding of the probability of some response for a given dose is only improved when we test at exactly that dose. Therefore, these are uncorrelated beta distributions.
**Algorithm 1.** Update rule for uncorrelated Beta distributions
This rule is for updating the beta distribution for the probability of observing response r for some dose $d_{i}$ based on the $\left(n+1\right)th$ data point. Let this $\left(n+1\right)th$ data point had dose $d_{j}$.BEGIN ALGORITHMIf $d_{i}=d_{j}$  If response r was observed for individual n+1   Set $\alpha_{i,r}^{n+1}=\alpha_{i,r}^{n}+1$   Set $\beta_{i,r}^{n+1}=\beta_{i,r}^{n}$  Else (response r was not observed for individual n+1)   Set $\alpha_{i,r}^{n+1}=\alpha_{i,r}^{n}$   Set $\beta_{i,r}^{n+1}=\beta_{i,r}^{n}+1$Else ($d_{i}\neq d_{j})$   Set $\alpha_{i,r}^{n+1}=\alpha_{i,r}^{n}$   Set $\beta_{i,r}^{n+1}=\beta_{i,r}^{n}$END ALGORITHM

##### Priors and Uninformative Priors

Initial values of α and β must be chosen. Typically, if there exists no prior knowledge for which response probabilities are most reasonable, it is best to use an uninformative prior. For this, the initial values of α and β for each dose $d_{i}\;$ for response r are set to 1 [28]. That is
(4)$$\alpha_{i,r}^{0}=1$$
(5)$$\beta_{i,r}^{0}=1$$

This is typically a reasonable choice, as prior to data being collected this leads to equal probability for each possible value of $p_{i,r}^{0}$ (Figure 2a). If there if is prior understanding about the probability of response r for dose $d_{i}$ alternative values of $\alpha_{i,r}^{0}$ and $\beta_{i,r}^{0}$ can be used. A method for choosing these is discussed below under the heading of ‘Utilising Expert Knowledge to Inform Continuous Correlated Beta Process Model Priors’ of this section and the implications of this when conducting dose-finding trials investigated in objective 4.

##### Kernel Functions

Above we noted that uncorrelated beta distributions do not allow for information about response probabilities from one dose to inform understanding of response probability at any other dose. The CCBP model allows information about response probability for one dose to inform understanding of response probability for ‘similar’ doses. We describe what it means for doses to be ‘similar’ using a similarity function, $K\left(d_{i},d_{j}\right)$, traditionally called a ‘kernel’ function. This is a function that takes two doses as input and returns a number between 0 and 1 that represents how similar those doses are.

In the context of vaccine dose optimisation, a kernel function $K\left(d_{i},d_{j}\right)$ follows these rules for all doses $d_{i}$ and $d_{j}$ in the dosing domain:(6)$$0\leq K\left(d_{i},d_{j}\right)\leq 1$$
(7)$$K\left(d_{i},d_{j}\right)=1\;\mathrm{if}\;\mathrm{and}\;\mathrm{only}\;\mathrm{if}\;\mathrm{the}\;\mathrm{doses}\;\mathrm{are}\;\mathrm{completely}\;\mathrm{similar}$$
(8)$$K\left(d_{i},d_{j}\right)=0\;\mathrm{if}\;\mathrm{and}\;\mathrm{only}\;\mathrm{if}\;\mathrm{the}\;\mathrm{doses}\;\mathrm{are}\;\mathrm{completely}\;\mathrm{dissimilar}$$
(9)$$K\left(d_{i},d_{j}\right)=K\left(d_{j},d_{i}\right),\;\mathrm{so}\;\mathrm{similarity}\;\mathrm{is}\;\mathrm{symmetrical}$$
(10)$$K\left(d_{i},d_{i}\right)=1,\;\mathrm{so}\;a\;\mathrm{dose}\;\mathrm{must}\;\mathrm{be}\;\mathrm{completely}\;\mathrm{self}-\mathrm{similar}$$
where ‘complete similarity’ would imply that clinicians/modellers believe that observing response/non-response for dose $d_{i}$ is equivalent to observing a response/non-response for dose $d_{j}$ for the purposes of predicting response probability for dose $d_{j}$. Likewise, ‘completely dissimilar’ would imply that clinicians/modellers do not believe that observing a response/non-response for dose $d_{i}$ provides any information regarding response probability for dose $d_{j}$.

We can then use kernels to inform beta distributions for multiple ‘similar’ doses based on data gathered for a specific dose. Using a kernel function makes these beta processes ‘correlated beta processes’.

The beta distribution update rule described in Algorithm 1 is then changed to that showed in Algorithm 2.
**Algorithm 2.** (Continuous) Correlated Beta Process Update Rule
This rule is for updating the beta distribution for the probability of observing response r for some dose $d_{i}$ based on the $\left(n+1\right)th$ data point. Let this $\left(n+1\right)th$ data point have been at dose $d_{j}$.BEGIN ALGORITHMCalculate $K\left(d_{i},d_{j}\right)$If response r was observed for individual n+1   Set $\alpha_{i,r}^{n+1}=\alpha_{i,r}^{n}+K\left(d_{i},d_{j}\right)$   Set $\beta_{i,r}^{n+1}=\beta_{i,r}^{n}$Else (response r was not observed for individual n+1)   Set $\alpha_{i,r}^{n+1}=\alpha_{i,r}^{n}$   Set $\beta_{i,r}^{n+1}=\beta_{i,r}^{n}+K\left(d_{i},d_{j}\right)$END ALGORITHM

An example of this update process is now given. Say doses, $d_{1}$ and $d_{2}$, have efficacy probabilities initially described by a flat prior, that is;
(11)$$p_{1,eff}^{0}\sim Beta\left(\alpha_{1,eff}^{0},\beta_{1,eff}^{0}\right)=Beta\left(1,1\right),$$
(12)$$p_{2,eff}^{0}\sim Beta\left(\alpha_{2,eff}^{0},\beta_{2,eff}^{0}\right)=Beta\left(1,1\right),$$

Say dose $d_{1}$ is tested and a positive efficacy response observed. If $d_{1}$ and $d_{2}$ are 50% similar (efficacy kernel $K\left(d_{1},d_{2}\right)$ = $K\left(d_{2},d_{1}\right)$ = 0.5), then
(13)$$p_{1,eff}^{1}\sim Beta\left(\alpha_{1,eff}^{0}+K\left(d_{1},d_{1}\right),\beta_{1,eff}^{0}\right)=Beta\left(1+1,\;1\right)=Beta\left(2,\;1\right),$$
(14)$$p_{2,eff}^{1}\sim Beta\left(\alpha_{2,eff}^{0}+K\left(d_{1},d_{2}\right),\beta_{2,eff}^{0}\right)=Beta\left(1+0.5,\;1\right)=Beta\left(1.5,\;1\right),$$

In this work, we chose to use the squared exponential kernel suggested in [27,29] defined as
(15)$$K\left(d_{i},d_{j}\right)=e^{-\frac{{\left(d_{i}-d_{j}\right)}^{2}}{l^{2}}}$$
where l is a length hyperparameter that can be chosen to adjust the range for which doses are considered similar (examples in Figure 4). For small l, the data only influences model prediction near the tested dose (Figure 4). For larger l, the data influences model prediction at a greater distance. In this work length parameter l = 0.2 for modelling single-administration vaccine dose–response [S2, S7.1].

This choice of kernel is means similarity is defined continuously for any two doses in the dosing domain, making these beta processes continuous correlated beta processes (CCBPs).

##### Modelling Prime/Boost Dose Response

Extending Continuous Correlated Beta Process (CCBP) models to modelling prime/boost dose response requires only a change to the kernel function. Since a squared exponential kernel was chosen, this change is intuitive. For doses $d_{i}$ and $d_{j}$, where dose $d_{i}$ has prime dose $d_{i1}$ and boost dose $d_{i2}$ the 2-dose kernel function would be
(16)$$K^{2}\left(d_{i},d_{j}\right)=e^{-\frac{{\left(d_{i1}-d_{j1}\right)}^{2}}{l_{1}^{2}}\;-\;\frac{{\left(d_{i2}-d_{j2}\right)}^{2}}{l_{2}^{2}}}$$

Similarly, for modelling prime/boost/second-boost dose response the 3-dose kernel function would be
(17)$$K^{3}\left(d_{i},d_{j}\right)=e^{-\frac{{\left(d_{i1}-d_{j1}\right)}^{2}}{l_{1}^{2}}\;-\;\frac{{\left(d_{i2}-d_{j2}\right)}^{2}}{l_{2}^{2}}\;-\;\frac{{\left(d_{i3}-d_{j3}\right)}^{2}}{l_{3}^{2}}}$$

This pattern can be generalised to considering to considering H doses
(18)$$K^{H}\left(d_{i},d_{j}\right)=e^{-\overset{H}{\underset{o=1}{\sum }}\frac{{\left(d_{io}-d_{jo}\right)}^{2}}{l_{o}^{2}}\;}$$

In this work the length parameters l=0.2 was used for modelling single-administration dose–response, $l_{1}=l_{2}=0.25$ were used for modelling prime/boost dose–response, and length parameters $l_{1}=l_{2}=l_{3}=0.4$ were used for modelling prime/boost/second-boost dose–response. Please refer to Appendix A for a description of how these values were chosen. Length parameters do not need to be equal but were equal here for simplicity.

##### Utilising Expert Know Ledge to Inform Continuous Correlated Beta Process Model Priors

It is possible that including expert knowledge into the modelling process may improve optimality of the final selected dose, leading to more effective early trial doses. Methods for including expert knowledge to inform the modelling process for parametric models have been previously discussed [41,42] but are non-trivial. Expert knowledge is integrated into the CCBP model by choosing different initial values for $\alpha_{i,r}^{0}$ and $\beta_{i,r}^{0}$ which we call the expert informed prior. For each dose $d_{i}$ and response r, an expert informed prior can be defined using the expert’s prediction of the most likely probability of response for that dose, $p_{i,r}^{expert}$, and level of confidence in that probability, $c_{i,r}^{expert}\geq 0$. These values could be based on previous knowledge of the vaccine or a similar product. $c_{i,r}^{expert}$ can be considered as the number of individuals worth of data that is required before the data influences the model prediction more than the expert knowledge. Incorporating expert priors in an initial Beta distribution for dose $d_{i}$ for response r is done by setting
(19)$$\alpha_{i,r}^{0}=p_{i,r}^{expert}\times c_{i,r}^{expert}+1,\;$$
(20)$$\beta_{i,r}^{0}=(1-p_{i,r}^{expert})\times c_{i,r}^{expert}+1$$

Then, the mode of the relevant Beta distribution for each $d_{i}$ will be $p_{i,r}^{expert}$ [43].

#### 2.2.2. Method of Trial DOSE Select Ion

The method for trial dose selection in the CoBe DOA is Thompson sampling. Thompson Sampling involves choosing clinical trial doses proportionally to the probability that they are optimal, given the available data and model. This is described in detail in [44,45,46,47], but the principle is to sample from Beta distributions for each dose, and then select the optimal dose based on the value of the utility function for each sample.

As a continuation of our earlier example, doses $d_{1}$ and $d_{2}$, had efficacy probabilities described, respectively, as
(21)$$p_{1,eff}^{1}\sim Beta\left(2,\;1\right),$$
(22)$$p_{2,eff}^{1}\sim Beta\left(1.5,\;1\right),$$
and we are using the maximum efficacy utility function. We can randomly sample efficacy probabilities ${\hat{p}}_{1,eff}$ and ${\hat{p}}_{2,eff}$ from these Beta distributions using statistical software. Then, the values of the utility function for $d_{1}$ and $d_{2}$ based on these samples are ${\hat{U}}_{1}={\hat{p}}_{1,eff}$ and ${\hat{U}}_{2}={\hat{p}}_{2,eff}$ If ${\hat{U}}_{1}>{\hat{U}}_{2}$ then $d_{1}$ is selected as the next dose to test, otherwise we select $d_{2}$. This process of sampling and selecting can be repeated to select as many trial doses as required for each sampling cohort (6 individuals per sampling cohort was used in this work).

#### 2.2.3. Method of Final Dose Selection

The method of final dose selection is to select the dose with maximum utility as given by the median probability prediction of response for each dose. For each dose $d_{i}$, the median probability of efficacy ${\bar{p}}_{i,eff}$ and ${\bar{p}}_{i,tox}$ are calculated. Then, a dose $d_{i}$ is predicted as optimal if $U\left({\bar{p}}_{i,eff},{\bar{p}}_{i,tox}\right)\geq U\left({\bar{p}}_{j,eff},{\bar{p}}_{j,tox}\right)\;$ for all doses dj, with ties broken randomly.

#### 2.2.4. Discretisation

Due to the continuous nature of the CCBP model, the CoBe dose optimisation approach can be applied when choosing between a potentially large number of doses, therefore the dosing domain can be discretised to include many doses.

#### 2.2.5. Full Correlated Beta (CoBe) Dose Optimisation Approach and an Example Trial

The complete CoBe dose optimisation approach is shown in Algorithm 3, and three time-points of an example simulated clinical trial is depicted in Figure 5. We note that, in practical application, clinicians/modellers may choose to skip step 7 of this algorithm until the final sampling cohort is completed. However, in this work this step was conducted after each cohort to investigate the effect of increasing trial size on improvement in vaccine dose selection.
**Algorithm 3.** Correlated Beta (CoBe) Dose Optimisation Algorithm

BEGIN ALGORITHM
1.Initialisation:
a.Choose in collaboration with clinicians and experts
i.Total trial participants available, Nii.Sampling cohort size, c (=6 in this work)iii.Determine whether a single-administration, prime/boost, or prime/boost/second-boost paradigm is being used.iv.Determine all potential doses, $d_{i}$, in the discretized dosing domain, see [Discretization]).v.Choose length parameter(s) for the efficacy similarity kernel (l=0.2, $l_{1}=l_{2}=0.25,\;\;l_{1}=l_{2}=l_{3}=0.4$ in this work)vi.Choose length parameter(s) for the toxicity similarity kernel (the same as for efficacy in this work)vii.Query experts to determine any potential priors.

2.Initialization of Beta distributions - in silico
a.Initialise description of efficacy probability distribution for each dose $d_{i}$ as $p_{i,eff}^{0}\sim Beta\left(\alpha_{i,eff}^{0},\beta_{i,eff}^{0}\right)$b.Initialise description of toxicity probability distribution for each dose $d_{i}$ as $p_{i,tox}^{0}\sim Beta\left(\alpha_{i,tox}^{0},\beta_{i,tox}^{0}\right)$
3.Thompson sampling for dose selection-in silico
a.For each dose $d_{i}$, sample ${\hat{p}}_{1,eff}$ and ${\hat{p}}_{2,eff}$ from the relevant Beta distributions.b.Select for trialing dose $d_{i}$ such that ${\hat{U}}_{i}>{\hat{U}}_{j}$ for all doses $d_{j}$, where $U_{i}\left(p_{i,eff},p_{i,tox}\right)\;$ is the utility function to be maximised.
4.Repeat step 3 until sampling cohort is full (c repeats total)5.Trialing and data collection *– practical*
a.Conduct a trial of c individuals, respectively, at the c doses chosen in steps 3 and 4. This is simulated in this work but would be practical lab work in real life application.b.Record c data points consisting of {dose given, whether efficacy was observed, whether toxicity was observed}
6.Model Updating-in silico
a.Update $\alpha_{i,eff}^{n-c},\beta_{i,eff}^{n-c},\alpha_{i,tox}^{n-c},\beta_{i,tox}^{n-c}$ to $\alpha_{i,eff}^{n},\beta_{i,eff}^{n},\alpha_{i,tox}^{n},\beta_{i,tox}^{n}$ using:
i.Update $\alpha {\;}_{i,eff}^{n-c},\beta_{i,eff}^{n-c},\alpha_{i,tox}^{n-c},\beta_{i,tox}^{n-c}$ to $\alpha_{i,eff}^{n-c+1},\beta_{i,eff}^{n-c+1},\alpha_{i,tox}^{n-c+1},\beta_{i,tox}^{n-c+1}$ using Algorithm 2 with a data point gathered in step 5.ii.Repeat for all other data points gathered in step 5 (order does not matter)7.Prediction of optimal dose *–* in silico
a.For each dose di, calculate the median response probabilities ${\bar{p}}_{i,eff}$ and ${\bar{p}}_{i,tox}$b.The predicted optimal dose is $d_{i}$ such that $U\left({\bar{p}}_{i,eff},{\bar{p}}_{i,tox}\right)\geq U\left({\bar{p}}_{j,eff},{\bar{p}}_{j,tox}\right)$
where $U_{i}\left(p_{i,eff},p_{i,tox}\right)\;$ is the utility function to be maximised.
8.Repeat steps 3-7 until all N trial participants have been utilised.
END ALGORITHM

#### 2.2.6. Other Dose-Optimisation Approaches

We consider multiple other dose-optimisation approaches other than the CoBe Dosing approach. These were the ‘Parametric’, ‘Adaptive Naive’, and ‘Uniform Naive’ DOAs.

##### Parametric Dose-Optimisation Approach

The ‘Parametric’ DOA uses parametric models to describe dose-efficacy and dose-toxicity, as described in [21,35]. Specifically, we used the latent quadratic model [35,48] for modelling dose-efficacy for single-administration dose-optimisation problems. This is given by
(23)$$p_{eff}\left(d_{i}\right)=latentquadratic\left(d_{i}\right)=logit\left(a+bd_{i}-cd_{i}^{2}\right),$$
with
(24)$$logit\left(z\right)=\frac{1}{1+e^{-z}}$$

Furthermore, parameters a,b,c.

We extended this model for prime/boost and prime/boost/second-boost dose-optimisation problems, respectively as
(25)$$p_{eff}\left(d_{i}\right)=latentquadratic_{2D}\left(d_{i}\right)=logit\left(a+b_{1}d_{i1}-c_{1}d_{i1}^{2}+b_{2}d_{i2}-c_{2}d_{i1}^{2}\right),$$
(26)$$p_{eff}\left(d_{i}\right)=latentquadratic_{3D}\left(d_{i}\right)=logit\left(a+b_{1}d_{i1}-c_{1}d_{i1}^{2}+b_{2}d_{i2}-c_{2}d_{i1}^{2}+b_{3}d_{i3}-c_{3}d_{i3}^{2}\right),$$
with parameters $a,b_{1},c_{1},b_{2},c_{2},\;b_{3}\;,c_{3}$ and dose $d_{i}$ having prime dose $d_{i1}$, boost dose $d_{i2}$, and potential second-boost dose $d_{i3}$. This is similar to the approach used in [49], but extended to allow for non-monotonicity in the dose-efficacy relationships.

We used the latent linear model [35,50] for modelling dose-toxicity for single-administration dose-optimisation problems. This is given as
(27)$$p_{tox}\left(d_{i}\right)=latentlinear\left(d_{i}\right)=logit\left(a+bd_{i}\right),$$
with parameters a,b

We similarly extended this model for prime/boost dose-optimisation problems, given as
(28)$$p_{tox}\left(d_{i}\right)=latentlinear_{2D}\left(d_{i}\right)=logit\left(a+b_{1}d_{i1}+b_{2}d_{i2}\right),$$
with parameters $a,b_{1},b_{2},$

These models can be calibrated to the available dose–response data by determining the maximum likelihood estimate of the parameters given the available data. Pseudo-data were used to aid stability of the model calibration, as described in both [30] and Appendix A. These calibrated models were then used to predict dose-utility. The method of trial dose selection for each cohort was the softmax selection method described in both [21,51,52] and Appendix A. The method of final dose selection was to choose the dose with the maximum utility according to the predictions of the calibrated model. Due to being a modelling method, for this DOA the discretized dosing domain could include a large number of potential doses.

##### Adaptive Naive Dose-Optimisation Approach

The ‘Adaptive Naive’ DOA has been well discussed in the past for conducting trials comparing treatments and doses [45,46]. Like the CoBe DOA, the probability of efficacy/toxicity for each potential dose is described by a beta distribution, the method of trial dose selection is Thompson sampling, and the method of final dose selection is the maximised median prediction. Unlike the CoBe DOA, however, this DOA does not make use of a similarity kernel or other modelling methods, so prediction of efficacy/toxicity for any given dose is determined by only considering previous data for that specific dose. Hence, this DOA is ‘adaptive’, but the predictions of efficacy/toxicity for any given dose are ‘naive’. Due to this lack of modelling, this DOA discretizes the dosing domain to only a small number of doses.

##### Uniform Naive Dose-Optimisation Approach

The ‘Uniform Naive’ DOA is perhaps the most common DOA used for selecting optimal vaccine dose, though it is typically not named as such. This is the same as the Adaptive Naive DOA, except that the method of trial dose selection is to divide all clinical trial participants evenly amongst the discretized dosing domain. Commonly all sampling cohorts would be conducted at the same time given that there is no adaptive design. There is no adaptive design or modelling, so this DOA is ‘uniform’ in its method of trial dose selection and ‘naive’ in its predictions of efficacy/toxicity for each dose. Again, like the Adaptive Naive DOA, this DOA discretizes the dosing domain to only a small number of doses.

### 2.3. Section 3. Definition of the Simulation Study Methodology and Details of the Implementation of This Methodology

#### 2.3.1. Definition of a Simulation Study

When conducting real life dose-finding studies we have the capacity to generate data through practical experiments. Trial individuals can be given vaccine doses, immunological/toxicity data responses recorded, and then these data can be used to make decisions regarding continued trial dose according to the DOA that is being used. In simulation studies we mimic this process, simulating clinical trial data generation according to ‘true’ vaccine dose-efficacy/dose-toxicity curves that we have defined and are hence known. We can define various different ‘true’ underlying dose–response curves to define different ‘scenarios’, which in turn allow the theoretical capacity of effective dose optimisation to be evaluated.

#### 2.3.2. Definition of a Scenario

In this work, a scenario consisted of:A dosing domain: Whether these scenarios consider a single dose or combinations of doses, and the range for which possible doses that could be tested or predicted as optimal, as described above. For simplicity, we considered that doses of vaccine (whether single administration or prime dose, or a boost dose) to have been scaled to be between 0 and 1, as described in both [53,54,55] and Appendix A. Thus, a zero dose does not necessarily correspond to no vaccine being given, but instead corresponds to the smallest dose that clinicians/developers may be willing to consider. This scaling was purely for convenience.A utility function: To weigh the relative benefit of efficacy, toxicity, or any other dose related outcome a utility function is needed. For this work we use either the ‘maximum efficacy’ or ‘utility contour’ utility functions defined in Section 2.1.1.Efficacy probabilities for all possible doses: For each dose in the dosing domain, there was some true probability of efficacy for each dose that was defined for the scenario.Toxicity probabilities for all possible doses: If our aim was to minimise toxicity as well as to maximise efficacy, as in the ‘utility contour’ utility function, there was some true probability of toxicity for each dose in the dosing domain that was defined for the scenario.

For details of scenario creation see S6.

#### 2.3.3. Simulation Study Parameters

##### Discretisation

Specifically in this work,

For all scenarios involving single-administration paradigm vaccine dose–response, for the CoBe and Parametric DOAs we discretized the dosing domain to 101 doses (0.00, 0.01, 0.02, …, 0.99, 1.00) and for the Adaptive Naive and Uniform Naive DOAs we discretised the dosing domain to 6 doses (0.0, 0.2, 0.4, 0.6, 0.8, 1.0).For all scenarios involving prime/boost paradigm vaccine dose response, for the CoBe and Parametric DOAs we discretized the dosing domain to 411 doses (a 21-by-21 grid of (0.00, 0.05, …, 0.95, 1.00)) and for the Adaptive Naive and Uniform Naive DOAs we discretised the dosing domain to 9 doses (a 3-by-3 grid of (0.0, 0.5, 1.0).For the scenario involving prime/boost/second-boost paradigm vaccine dose response, for the CoBe and Parametric DOAs we discretized the dosing domain to 1331 doses (an 11-by-11-by-11 grid of (0.00, 0.10, …, 0.90, 1.00)) and for the Adaptive Naive and Uniform Naive DOAs we discretised the dosing domain to 27 doses (a 3-by-3-by-3 grid of (0.0, 0.5, 1.0).

##### Trial Size/Sampling Cohort Size

As the number of individuals available for conducting a dose-ranging trial may vary in real life, we assessed the performance of the DOAs for trial sizes from 6 to 300 individuals. These are sizes reasonable given typical phase I/early phase II vaccine trial sizes [9].

Additionally, as we consider adaptive DOAs, we had to specify the size of the sampling cohort. This was the number of individuals that were tested in each round of modelling/trialing (CoBe Dose Optimisation Approach Algorithm box steps 4, 5). The CoBe, Parametric, and Adaptive Naive DOAs used a sampling cohort size of six in this work for all scenarios. The Uniform Naive DOA used a sampling cohort size equal to the number of doses in the discretised dosing domain (either 6, 9 or 27).

#### 2.3.4. Metrics to Evaluate Dose-Optimisation Approaches

We used two metrics to evaluate the DOAs described in this work; one was related to optimal dose selection and one related to ethical trial design. Either of these may be considered to be important considerations for conducting vaccine dose-ranging trials.

True efficacy/utility of predicted optimal dose: After each cycle of trial/modelling (each sampling cohort), each DOA can recommend a dose that is predicted optimal given the current data. As this was a simulation study, we were aware of the true efficacy/utility at that selected dose. This true efficacy/utility of the selected doses was averaged across trial simulations to assess the ability of a dose finding approach to locate optimal dose.Cumulative sum of efficacy/utility: Each individual in a trial may have an efficacious response and may experience vaccine-related toxicity. The cumulative number of efficacious responses (or cumulative utility if both efficacy and toxicity are being optimised for) was averaged across simulations to assess the ability of a dose finding approach to maximise trial efficacy/utility.

The formula for ‘cumulative sum of efficacy’ after the first *n* individuals was
(29)$$cumulative\;efficacy\left(n\right)=count_{efficacy}\left(n\right)$$
and the formula for ‘cumulative sum of utility’ after the first *n* individuals was
(30)$$cumulative\;utility\left(n\right)=n\times U\left(\frac{count_{efficacy}\left(n\right)}{n},\frac{count_{toxicity}\left(n\right)}{n}\right)$$
with U() being the utility contour function defined in Section 1 and $count_{efficacy}\left(n\right)/count_{toxicity}\left(n\right)$ being the number of individuals that had experienced efficacy/toxicity in the first *n* individuals.

#### 2.3.5. Implementation

Each scenario/approach pairing was simulated 100 times. The mean for both evaluation metrics for these 100 simulated clinical trials was calculated.

The simulation study was conducted in Python, using SciPy for statistical inference, for implementation of Beta distributions, and for calibration of the parametric models for the Parametric DOA [56].

### 2.4. Section 4. Description of the Use of the Concepts Defined above in Evaluating the Correlated Beta Dose-Optimsation Approach in the Context of Our Objectives

#### 2.4.1. Objective 1. Evaluate the Correlated Beta Dose Optimisation Approach for Optimising Vaccine Efficacy for a Single Dose Administration

In this objective we aimed to evaluate the vaccine dose-optimisation ability of the CoBe DOA when compared to other DOAs when choosing a single administration dose that maximises efficacy. Using the definition of scenarios above, we consider scenarios with:Dosing domain: Single-administrationUtility function: Maximise EfficacyEfficacy curve: Is defined for each scenarioToxicity curve: Not defined/not of interest

Seven scenarios were used to explore this objective (Figure 6). These scenarios reflect cases for which vaccine dose efficacy may be
5.gently saturating (Figure 6a)6.sharply saturating (Figure 6b)7.gently peaking (Figure 6c)8.sharply peaking (Figure 6d)9.decreasing (Figure 6e)10.undulating (Figure 6f)11.flattened peaking (Figure 6g)

For each scenario, we simulated 100 clinical trials conducted under each of the four DOAs (100 × 7 × 4 simulated trials total). Cohorts were of size 6, and we simulated 50 cohorts for each clinical trial (300 total individuals per simulated trial). For each clinical trial, after each cohort an optimal dose was predicted and used to calculate ‘true efficacy at predicted optimal dose’.

The cumulative number of efficacious responses that had occurred up to and including that cohort in the simulated trials was also calculated as the ‘cumulative sum of efficacy’. The mean value of these two metrics after each cohort across the hundred simulations were then calculated for each scenario for each DOA. A 95% confidence interval for the true mean values of these metrics were also calculated. These were plotted to qualitatively show the expected ‘true efficacy at the predicted optimal’ and ‘cumulative sum of efficacy’ for each scenario for clinical trials conducted using each DOA after each cohort. This allowed comparison between the DOAs, and also a comparison to the theoretical true optimal that a DOA could achieve.

#### 2.4.2. Objective 2: Evaluate the C orrelated Beta Dose Optimisation Approach for Optimising Vaccine Efficacy for a Prime-Dose/Boost-Dose Administration

In this objective we aim to assess the dose-optimisation ability of the CoBe dose-optimisation approach compared to other dose-optimisation approaches when choosing doses for a multiple administration vaccine that maximises efficacy. In this objective, we use scenarios where:Dosing domain: Prime/boost (scenarios 1–5) or prime/boost/second-boost (scenarios 6,7)Utility function: Maximise EfficacyEfficacy curve: Is defined for each scenarioToxicity curve: Not defined/not of interest

Seven scenarios were used to explore this objective (Figure 5). The prime/boost scenarios reflect cases for which vaccine dose efficacy may be

12.Peaking with respect to both doses and where the combination of both vaccine doses increases their efficacy (Figure 7a)13.Saturating with regard to both doses but where the combination of both vaccine doses decreases their efficacy (Figure 7b)14.Saturating with respect to both doses and where the combination of both vaccine doses increases their efficacy (Figure 7c)15.Saturating with respect to both doses and where the combination of both vaccines increases their efficacy, but maximally dosing both vaccines causes decreased efficacy. (Figure 7d)16.Saturating with respect to both doses and where the combination of both vaccines increases their efficacy, but where one of the doses is significantly more important to maximising efficacy. (Figure 7e)Scenarios 6 and 7 are prime/boost/second-boost. These scenarios:17.Represents a case where there is a maximally efficacious dose for each, and any increase/decrease in any of these doses decreases efficacy regardless of the other doses. Thus, the optimal dose for each of the prime/boost/second-boost was independent of what other doses were selected (Figure 7f)18.Represent a case where a maximal dose of any two of the three doses produces a highly efficacious response, but a maximal dose of all three does not produce a highly efficacious response (Figure 7g).

For each scenario, we simulated 100 clinical trials conducted under each of the four DOAs (100 × 7 × 4 simulated trials total). For the CoBe, Parametric and Adaptive Naive DOAs, cohorts were of size 6, and we simulated 50 cohorts for each clinical trial (300 total individuals per simulated trial). For scenarios 1–5, the Uniform Naive DOA used cohorts of size 9, and we simulated 33 cohorts for each clinical trial (297 total individuals per simulated trial). For scenarios 6 and 7, the Uniform Naive DOA used cohorts of size 27, and we simulated 11 cohorts for each clinical trial (297 total individuals per simulated trial).

For each clinical trial, after each cohort an optimal dose was predicted and used to calculate ‘true efficacy at predicted optimal dose’. The cumulative number of efficacious responses that had occurred up to and including that cohort in the simulated trials was also calculated as the ‘cumulative sum of efficacy’. The mean value of these two metrics after each cohort across the hundred simulations were then calculated for each scenario for each DOA. A 95% confidence interval for the true mean values of these metrics were also calculated. These were plotted to qualitatively show the expected ‘true efficacy at the predicted optimal’ and ‘cumulative sum of efficacy’ for each scenario for clinical trials conducted using each DOA after each cohort. This allowed comparison between the DOAs, and also a comparison to the theoretical true optimal that a DOA could achieve.

#### 2.4.3. Objective 3. Evaluate the Correlated Beta Dose Optimisation Approach for Optimising Vaccine Utility, Maximising Efficacy, and Minimising Toxicity

In this objective we aim to assess the dose-optimisation ability of the CoBe dose-optimisation approach compared to dose-optimisation approaches when choosing doses for single or multiple administration vaccines for which an optimal balance of efficacy and toxicity must be achieved. In this objective we use scenarios where:Dosing domain: Single-administration (scenarios 1–4) or prime/boost administration (scenarios 5–6)Utility function: Utility ContourEfficacy curve: Is defined for each scenarioToxicity curve: Is defined for each scenario

Six scenarios were used in this objective (Figure 8). The single-administration scenarios reflect cases for which vaccines

19.have gradually increasing efficacy and toxicity with dose (Figure 8a–c)20.have sharply peaking efficacy and gradually increasing toxicity with dose (Figure 8 d–f)21.have gradually increasing efficacy and sharply increasing toxicity with dose (Figure 8g–i)22.have sharply peaking efficacy and sharply increasing toxicity with dose (Figure 8j–l)The prime/boost administration scenarios reflect cases for which vaccines:23.have efficacy as per objective 2, scenario 3, and toxicity increasing for high doses of either vaccine (Figure 8m–o)24.have efficacy as per objective 2, scenario 2, and toxicity increasing for high doses of either vaccine (Figure 8p–r)

For each scenario, we simulated 100 clinical trials conducted under each of the four DOAs (100 × 6 × 4 simulated trials total). For the CoBe, Parametric and Adaptive Naive DOAs, cohorts were of size 6, and we simulated 50 cohorts for each clinical trial (300 total individuals per simulated trial). This was also true for the Uniform Naive DOA for scenarios 1–4. For scenarios 5 and 6, the Uniform Naive DOA used cohorts of size 9, and we simulated 33 cohorts for each clinical trial (297 total individuals per simulated trial).

For each clinical trial, after each cohort an optimal dose was predicted and used to calculate ‘true utility at predicted optimal dose’. The cumulative utility up to and including that cohort in the simulated trials was also calculated as the ‘cumulative sum of utility’. The mean value of these two metrics after each cohort across the hundred simulations were then calculated for each scenario for each DOA. A 95% confidence interval for the true mean values of these metrics were also calculated. These were plotted to qualitatively show the expected ‘true utility at the predicted optimal’ and ‘cumulative sum of utility’ for each scenario for clinical trials conducted using each DOA after each cohort. This allowed comparison between the DOAs, and also a comparison to the theoretical true optimal that a DOA could achieve.

#### 2.4.4. Objective 4. Evaluate the Use of Expert Knowledge Informed Continuous Correlated Beta Process Priors for Vaccine Dose-Optimisation

In this objective we aim to assess the dose-optimisation ability of the CoBe dose-optimisation approach when including either accurate or inaccurate expert information priors, and to what extent such priors improve or are detrimental to CoBe DOA performance.

We compared the CoBe DOA with 5 different ‘priors’Very strong, correctStrong, correctNo priorStrong, incorrectVery strong, incorrect

These priors were implemented as defined in 2.2.1.4. The ‘No prior’ approach had $p_{i,r}^{0}\sim Beta\left(\alpha_{i,eff}^{0},\beta_{1i,eff}^{0}\right)=Beta\left(1,1\right)$ for all doses and is the CoBe DOA used in the previous objectives. The ‘Very strong, correct’ and ‘Strong, correct’ priors assume the expert knowledge is entirely correct to the ‘true’ vaccine dose response, with suggested probability $p_{i,r}$ equal to the true probability of efficacy/toxicity for all doses $d_{i}$ (so if the true probability of efficacy for some dose was 0.2, the ‘expert’ would predict an efficacy of 0.2). The ‘Very strong, incorrect’ and ‘Strong, incorrect’ priors represent the expert being largely incorrect, with suggested probability $p_{i,r}$ equal to one minus the true probability of efficacy/toxicity for all doses (so if the true probability of efficacy for some dose was 0.2, the ‘expert’ would predict an efficacy of 0.8). The ‘Strong, correct’ and ‘Strong, incorrect’ priors used $c_{i}=3$, which was deemed appropriate based on previous results in parametric dose-optimisation [57,58] and the ‘Very strong, correct’ and ‘Very strong, incorrect’ priors used $c_{i}=20\;$ for all doses which represented having extreme confidence in the expert prior.

In this objective we use scenarios from the above 3 objectives (Figure 9). In this objective we use scenarios where:Dosing domain: Single-administration (scenarios 1, 2), prime/boost administration (scenarios 3,4, 7), or prime/boost/second-boost administration (scenarios 5,6)Utility function: Maximise Efficacy (Scenarios 1–6), Utility Contour (Scenario 7)Efficacy curve: Is defined for each scenarioToxicity curve: Is defined for only scenario 7

Specifically, these are:25.Objective 1, Scenario 1 (Figure 9a)26.Objective 1, Scenario 4 (Figure 9b)27.Objective 2, Scenario 1 (Figure 9c)28.Objective 2, Scenario 2 (Figure 9d)29.Objective 2, Scenario 6 (Figure 9e)30.Objective 2, Scenario 7 (Figure 9f)31.Objective 3, Scenario 6 (Figure 9g–i)

For each scenario, we simulated 100 clinical trials conducted under each of the four DOAs (100 × 7 × 4 simulated trials total). Cohorts were of size 6, and we simulated 50 cohorts for each clinical trial (300 total individuals per simulated trial).

For each clinical trial, after each cohort an optimal dose was predicted and used to calculate ‘true efficacy at predicted optimal’ for scenarios 1–6 and ‘true utility at predicted optimal dose’ for scenario 7. The cumulative efficacy utility up to and including that cohort in the simulated trials was also calculated as the ‘cumulative sum of efficacy’ for scenarios 1–6, similarly the ‘cumulative sum of utility’ was calculated for scenario 7. The mean value of these metrics after each cohort across the hundred simulations were then calculated for each scenario for each DOA. A 95% confidence interval for the true mean values of these metrics were also calculated. These were plotted to qualitatively show the expected ‘true efficacy/utility at the predicted optimal’ and ‘cumulative sum of efficacy/utility’ for each scenario for clinical trials conducted using each DOA after each cohort. This allowed comparison between the DOAs, and a comparison to the theoretical true optimal that a DOA could achieve.

## 3. Results

### 3.1. Objective 1. Evaluate the Correlated Beta Dose Optimisation Approach for Optimising Vaccine Efficacy for a Single Dose Administration

#### 3.1.1. True Efficacy at Predicted Optimal Dose

The DOA (dose-optimisation approach) that best maximised ‘true efficacy at predicted optimal dose’ (from here called ‘true efficacy’) for a given scenario and trial size was considered to be the ‘best’ DOA for the aim of selecting an optimal dose for that scenario. The left-hand side of Figure 10 shows the change in mean true efficacy with increasing numbers of trial participants for each of the four DOAs for each scenario, averaged across 100 simulated clinical trials. For each of these plots the upper and lower brown lines, respectively, show the maximal and minimal efficacy possible for that scenario. A mean true efficacy for a DOA being closer to the upper brown line relative to a second DOA indicates the first DOA being on average better at selecting a highly efficacious dose. Equivalently, a mean true efficacy being closer to the lower brown line would represent a DOA being of average worse at selecting a highly efficacious dose.

For all scenarios, the CoBe (Correlated Beta) DOA had similar or greater mean true efficacy relative to the other DOAs for most trial sizes (Figure 10, left-hand side (LHS)). The mean true efficacy curve typically plateaued with between 30 and 90 trial participants. Scenario 7 was an exception, which may be due to the flat nature of the scenario’s dose-efficacy curve, which may have necessitated more data in order to discern a statistical difference in dose-efficacy for different doses (note the scale of the *y*-axis).

The CoBe DOA had a lower/worse mean true efficacy than the Parametric DOA for scenarios 1 and 2 for small numbers of trial participants. This may have been because the parametric model for the Parametric DOA was able to easily approximate the scenario 1 and 2 dose response curves with minimal data and correctly identify that the largest dose was maximally efficacious. However, the CoBe DOA had a greater mean true efficacy than the Parametric DOA for scenarios 4 and 6 for most trial sizes.

The CoBe DOA had similar mean true efficacy to the Adaptive Naive DOA for most scenarios. However, for scenarios 3 and 6 the Adaptive Naive DOA plateaued at a lower true efficacy. For scenarios where one of the potential doses in the discretised dosing domain of the Adaptive Naive DOA was near optimal (1,2,4,5 and 7), the Adaptive DOA had similar mean true efficacy to the CoBe DOA. However, when none of these potential doses were near optimal, the Adaptive Naive DOA had a lower/worse mean true efficacy than the CoBe DOA. For example, for scenarios 3 and 6, none of the six doses in the discretised dosing domain were truly optimal for those scenarios.

The uniform naive approach had a lower/worse mean true efficacy compared to the other approaches, particularly when the number of trial participants was small. The only exception was that the mean true efficacy for the Uniform Naive DOA was greater than that of the Parametric DOA for scenario 6. The mean true efficacy of the Uniform Naive DOA was typically equaled or surpassed by that of the Adaptive Naive DOA for all scenarios and numbers of trial participants.

#### 3.1.2. Cumulative Sum of Efficacy

Cumulative sum of efficacy measures a DOA’s capacity to maximise the benefit to trial participants. After a certain number of trial participants, a DOA with a higher cumulative efficacy would be considered ‘more ethical’ than a DOA with a lower cumulative efficacy, as it would reflect those trial participants having received on average more efficacious dosing. The right-hand side (RHS) of Figure 10 shows the change in mean cumulative sum of efficacy with increasing numbers of trial participants for each of the four DOAs for each scenario, averaged across 100 simulated clinical trials. For each of these plots the upper and lower brown lines, respectively, show the theoretical maximal and minimal mean cumulative efficacy possible for that scenario. A mean true efficacy for a DOA being closer to the upper brown line relative to a second DOA reflects that the trial participants for simulated clinical trials using the first DOA on average received more efficacious doses. If the relationship between number of trial participants and mean cumulative sum of efficacy for a DOA becomes parallel to the upper brown line after some number of trial participants, then that represents that the DOA gave a near optimal dose to all trial participants after that point. No DOA could exceed this upper brown line, as this upper brown line reflects a DOA for which every trial participant receives the dose that is truly most efficacious.

The relationship between the number of trial participants and cumulative sum of efficacy for DOAs that used adaptive trial design (CoBe, Parametric, Adaptive Naive DOAs) was non-linear (Figure 10, RHS). For small numbers of trial participants, the gradient of this curve was less steep than the line for the theoretical maximal (upper brown line, Figure 10 RHS). When the number of trial participants was small, there was little data available to guide the adaptive design of these DOAs and so trial doses were more likely to be suboptimal. As the number of trial participants increased, more data was available to inform selection of trial doses that were likely to be efficacious, increasing the steepness of the curve.

The CoBe DOA typically had a similar or greater cumulative sum of efficacy relative to the other DOAs for all scenarios (Figure 10, RHS). In many scenarios the mean cumulative efficacy was only slightly below the theoretical maximum for that scenario for the number of trial participants. Given that this upper bound is only achievable by dosing all trial participants at the true optimal dose for that scenario (which is likely not known a priori in a dose-finding trial), for these scenarios the CoBe DOA was found to be highly ethical. The Uniform Naive DOA had a lower/worse cumulative sum of efficacy than the other DOAs for all scenarios, especially at large trial size. This was likely because this DOA did not use any adaptive design, which is what allowed other DOAs to choose more promising doses for their later trial participants.

### 3.2. Objective 2. Evaluate the Correlated Beta Dose Optimisation Approach for Optimising Vaccine Efficacy for a Prime-Dose/Boost-Dose Administration

#### 3.2.1. True Efficacy at Predicted Optimal Dose

The DOA that best maximised true efficacy for a given scenario and trial size was considered to be the ‘best’ DOA for the aim of selecting an optimal dose for that scenario. The left-hand side of Figure 11 shows the change in mean true efficacy with increasing numbers of trial participants for each of the four DOAs for each scenario, averaged across 100 simulated clinical trials. For each of these plots the upper and lower brown lines, respectively, show the maximal and minimal efficacy possible for that scenario. A mean true efficacy for a DOA being closer to the upper brown line relative to a second DOA indicates the first DOA being on average better at selecting a highly efficacious dose. Equivalently, a mean true efficacy being closer to the lower brown line would represent a DOA being of average worse at selecting a highly efficacious dose.

For all scenarios, the CoBe (Correlated Beta) DOA had similar or greater mean true efficacy relative to the other DOAs for most trial sizes (Figure 11, LHS). The mean true efficacy curve typically plateaued with between 60 and 90 trial participants.

The CoBe DOA had a lower/worse mean true efficacy than the Parametric DOA for scenarios 3 and 5 for small numbers of trial participants but had a higher mean true efficacy than the Parametric DOA for scenarios 2 and 7.

The CoBe DOA had a higher mean true efficacy than the Adaptive Naive DOA for scenarios 4 and 6 and for a small numbers of trial participants for scenario 7 and a large number of trial participants for scenario 1. Otherwise mean true efficacy was similar for these two DOAs.

Again, the Uniform Naive DOA had a lower/worse mean true efficacy than all other approaches for all scenarios, with the only exception being the Parametric DOA which had a lower mean true efficacy for scenarios 2 and 7.

#### 3.2.2. Cumulative Sum of Efficacy

Cumulative sum of efficacy measures a DOA’s capacity to maximise the benefit to trial participants. After a certain number of trial participants, a DOA with a higher cumulative efficacy would be considered ‘more ethical’ than a DOA with a lower cumulative efficacy, as it would reflect those trial participants having received on average more efficacious dosing. The right-hand side of Figure 11 shows the change in mean cumulative sum of efficacy with increasing numbers of trial participants for each of the four DOAs for each scenario, averaged across 100 simulated clinical trials. For each of these plots the upper and lower brown lines, respectively, show the theoretical maximal and minimal mean cumulative efficacy possible for that scenario. A mean true efficacy for a DOA being closer to the upper brown line relative to a second DOA reflects that the trial participants for simulated clinical trials using the first DOA on average received more efficacious doses. If the relationship between number of trial participants and mean cumulative sum of efficacy for a DOA becomes parallel to the upper brown line after some number of trial participants, then that represents that the DOA gave a near optimal dose to all trial participants after that point. No DOA could exceed this upper brown line, as this upper brown line reflects a DOA for which every trial participant receives the dose that is truly most efficacious.

The same non-linearity in the relationship between number of trial participants and cumulative sum of efficacy for DOAs that used adaptive trial design (CoBe, Parametric, Adaptive Naive DOAs) that was observed in objective 1 was also observed in objective 2 (Figure 11, RHS)

The CoBe DOA typically had a similar or greater cumulative sum of efficacy relative to the other DOAs for all scenarios (Figure 11, RHS). The exception was for scenario 6, for which the Parametric DOA had a greater cumulative sum of efficacy, however the CoBe DOA had a greater cumulative sum of efficacy for scenarios 2 and 7. The Adaptive Naive DOA had a greater cumulative sum of efficacy than the CoBe DOA for scenarios 2 and 6, however the CoBe DOA had a greater cumulative sum of efficacy for scenarios 4 and 7. The Uniform Naive DOA had a lower/worse cumulative sum of efficacy than the other DOAs for all scenarios, especially at large trial size.

### 3.3. Evaluate the Correlated Beta Dose Optimisation Approach for Optimising Vaccine Utility, Maximising Efficacy and Minimising Toxicity

#### 3.3.1. True Utility at Predicted Optimal Dose

The DOA that best maximised ‘true utility at predicted optimal dose’ (from here called ‘true utility’) for a given scenario and trial size was considered to be the ‘best’ DOA for the aim of selecting an optimal dose for that scenario. Utility was defined by the ‘utility contour’ function that increased with an increased probability of efficacy and increased with a decreased probability of toxicity. The left-hand side of Figure 12 shows the change in mean true utility with increasing numbers of trial participants for each of the four DOAs for each scenario, averaged across 100 simulated clinical trials. For each of these plots the upper and lower brown lines, respectively, show the maximal and minimal utility possible for that scenario. A mean true utility for a DOA being closer to the upper brown line relative to a second DOA indicates the first DOA being on average better at selecting a high utility dose. Equivalently, a mean true efficacy being closer to the lower brown line would represent a DOA being on average worse at selecting a high utility dose.

For all scenarios, the CoBe (Correlated Beta) DOA had similar or greater mean true utility relative to the other DOAs for all trial sizes (Figure 10, LHS). The mean true utility curve typically plateaued with between 30 and 60 trial participants for the single dose administration scenarios (1–4), and between 60 and 90 trial participants for the prime/boost administration (5 and 6).

The CoBe DOA had similar mean true utility to the Parametric DOA for scenarios 1–4, however for scenarios 5 and 6 the CoBe DOA had a greater mean true utility relative to the Parametric DOA. The CoBe DOA had a greater mean true utility than either of the Adaptive Naive or Uniform Naive DOAs for all scenarios.

#### 3.3.2. Cumulative Sum of Utility

Cumulative sum of utility measures a DOA’s capacity to maximise the benefit to trial participants. After a certain number of trial participants, a DOA with a higher cumulative utility would be considered ‘more ethical’ than a DOA with a lower cumulative utility, as it would reflect those trial participants having received on average more efficacious/less toxic dosing. The right-hand side of Figure 12 shows the change in mean cumulative utility with increasing numbers of trial participants for each of the four DOAs for each scenario, averaged across 100 simulated clinical trials. For each of these plots the upper and lower brown lines, respectively, show the theoretical maximal and minimal mean cumulative utility possible for that scenario. A mean true utility for a DOA being closer to the upper brown line relative to a second DOA reflects that the trial participants for simulated clinical trials using the first DOA on average received more efficacious/less toxic doses. If the relationship between number of trial participants and mean cumulative sum of utility for a DOA becomes parallel to the upper brown line after some number of trial participants, then that represents that the DOA gave a near optimal dose to all trial participants after that point. No DOA could exceed this upper brown line, as this upper brown line reflects a DOA for which every trial participant receives the dose that is truly optimal in regard to maximising efficacy whilst minimising toxicity according to the utility function.

The same non-linearity in the relationship between number of trial participants and cumulative utility for DOAs that used adaptive trial design (CoBe, Parametric, Adaptive Naive DOAs) that was observed in objectives 1 and 2 was also observed in objective 3 (Figure 11, RHS). The CoBe DOA had a similar mean cumulative utility to the Parametric DOA for scenarios 1, 5 and 6, however the Parametric DOA had a greater mean cumulative utility for scenarios 2,3 and 4. The CoBe DOA had a similar or greater mean cumulative utility to the Adaptive Naive DOA for all scenarios other than scenario 2. The Uniform Naive DOA had a lower/worse cumulative utility than the other DOAs for all scenarios, especially at large trial size.

### 3.4. Objective 4. Evaluate the Use of Expert Knowledge Informed Continuous Correlated Beta Process Priors for Vaccine Dose-Optimsation

In this objective we assessed how the CoBe DOA that was discussed in objectives 1–3 would be impacted by the inclusion of expert priors as discussed under the heading of ‘Utilising Expert Knowledge to Inform Continuous Correlated Beta Process Model Priors’ in Section 2.2.1. The black line in Figure 13 (‘No Prior’) reflects the CoBe DOA as it was used in the previous objectives.

#### 3.4.1. True Efficacy/Utility at Predicted Optimal Dose

The DOA that best maximised ‘true efficacy at predicted optimal dose’ or ‘true utility at predicted optimal dose’ (from here called ‘true efficacy/utility’) for a given scenario and trial size was considered to be the ‘best’ DOA for the aim of selecting an optimal dose for that scenario. The left-hand side of Figure 13 shows the change in mean true efficacy/utility with increasing numbers of trial participants for each of the five DOAs for each scenario, averaged across 100 simulated clinical trials. For each of these plots the upper and lower brown lines, respectively, show the maximal and minimal efficacy/utility possible for that scenario. A mean true efficacy/utility for a DOA being closer to the upper brown line relative to a second DOA indicates the first DOA being on average better at selecting a high efficacy/utility dose. Equivalently, a mean true efficacy/utility being closer to the lower brown line would represent a DOA being on average worse at selecting a high efficacy/utility dose.

The CoBe DOAs that used ‘Strong, Correct’ and ‘Very Strong, Correct’ CCBP (continuous correlated beta process) priors had greater mean true efficacy/utility than the ‘No Prior’ CoBe DOA for all scenarios. The CoBe DOAs that used ‘Strong, Incorrect’ and ‘Very Strong, Incorrect’ CCBP priors had lower/worse mean true efficacy/utility than the ‘No Prior’ CoBe DOA for all scenarios. For all scenarios other than scenario 1, the CoBe DOA with the ‘Very Strong, Incorrect’ prior failed to predict a near optimal dose, even for large numbers of trial participants.

For the ‘Very Strong, Correct’ and ‘Strong, Correct’ CCBP prior DOAs, the mean true efficacy/utility decreased with the number of trial participants for early cohorts. This was expected as the expert prior is correct, therefore the initial predicted optimal dose is truly optimal and true efficacy/utility could not increase relative to this.

#### 3.4.2. Cumulative Sum of Efficacy/Utility

Cumulative sum of efficacy/utility measures a DOA’s capacity to maximise the benefit to trial participants. After a certain number of trial participants, a DOA with a higher cumulative efficacy/utility would be considered ‘more ethical’ than a DOA with a lower cumulative efficacy/utility The right-hand side of Figure 13 shows the change in mean cumulative utility with increasing numbers of trial participants for the CoBe DOAs with each of the five types of CCBP prior for each scenario, averaged across 100 simulated clinical trials. For each of these plots the upper and lower brown lines, respectively, show the theoretical maximal and minimal mean cumulative efficacy/utility possible for that scenario. A mean true efficacy/utility for a DOA being closer to the upper brown line relative to a second DOA reflects that the trial participants for simulated clinical trials using the first DOA on average received more efficacious or more efficacious/less toxic doses. No DOA could exceed this upper brown line, as this upper brown line reflects a DOA for which every trial participant receives the dose that is truly optimal.

The CoBe DOAs with ‘Very Strong, Correct’ and ‘Strong, Correct’ CCBP priors had greater cumulative efficacy/utility for all scenarios relative to the ‘No Prior’ CoBe DOA (Figure 13, RHS). The CoBe DOAs with ‘Very Strong, Incorrect’ and ‘Strong, Incorrect’ CCBP priors had worse cumulative efficacy/utility for all scenarios relative to the ‘No Prior’ CoBe DOA.

## 4. Discussion

In this work we used simulation studies to evaluate the Correlated Beta (CoBe) dose optimisation approach (DOA), a novel mathematical-modelling methodology for selecting optimal vaccine dose that makes use of the non-parametric Continuous Correlated Beta Process (CCBP) model. We found that the CoBe DOA is effective and ethical for finding a vaccine dose which maximises efficacy and minimises toxicity for both single-administration and prime/boost administration regimens. The CoBe DOA typically had similar or preferable capacity to select optimal vaccine dose with maximal benefit to trial participants when compared to other DOAs. Additionally, this DOA can be further improved if there is correct and informative prior expert knowledge regarding vaccine dose-efficacy curves. This work suggests that mathematical modelling and adaptive trial design can lead to better vaccine dosing strategies. Further to this, non-parametric models and specifically the non-parametric CCBP model might be particularly useful if an appropriate parametric model is unknown. This may allow for more practical application of modelling in vaccine dose selection, accelerating vaccine development and saving lives.

This work is novel within the field of mathematical modelling-based vaccine dose selection, and the CCBP has also not previously been investigated for its potential to aid in optimising vaccine dose. The context in which we evaluated the methods was broad, as we conducted simulation studies that included single and prime/boost-administration vaccine dose-optimisation, both efficacy maximisation and toxicity minimisation, and the impact of incorporating expert opinion and knowledge into the modelling process.

The CoBe DOA has several strengths relative to the other DOAs discussed in this work. The CCBP models that were used to predict vaccine dose–response make simpler assumptions and are simpler to implement and interpret compared to other parametric/non-parametric mathematical models. Further, CCBP models can be extended to modelling prime/boost dose–response with only minor alterations and can be modified to benefit from expert knowledge with similar ease. The simplicity of the CCBP model did not hinder its ability to predict optimal vaccine dose in this simulation study, with CoBe DOA appearing equivalent or better at predicting optimal dose relative to other common dose-optimisation approaches. As the CCBP model did not rely on any biologically based assumptions, it could likely be generalised for the purposes of optimising vaccine dose, time between doses, adjuvant dose, or many other parameters related to vaccine administration that could impact vaccine efficacy and utility.

Beyond introducing and evaluating this new DOA, this work has several other strengths.

Firstly, we evaluated multiple other DOAs as part of this analysis, which highlights the potential strengths or weaknesses of these DOAs relative to each other. This analysis was conducted over many simulated scenarios, which showed that for different ‘true’ dose-efficacy/dose-toxicity relationships the performance of DOAs can vary. This work is also novel in highlighting that the potential efficacy/utility of a vaccine may be limited if dose is selected using a DOA that only considers a small number of potential doses, such as the Adaptive Naive and Uniform Naive DOAs. For example, for many scenarios (objective 1 scenario 3, objective 2 scenarios 1,4, and 6, and all objective 3 scenarios), the Adaptive Naive and Uniform Naive DOAs failed to find the true optimal dose as none of the few doses that were considered by these DOAs were optimal for those scenarios.

Finally, this work showed further evidence that using mathematical modelling and/or adaptive design may be both more effective for selecting optimal vaccine dose and more ethical than the ‘Uniform Naive’ DOA, which is equivalent to the standard approach in vaccine dose ranging trials.

There were weaknesses with both this work and the CoBe DOA that we proposed.

Firstly, this work and its findings are based on simulated clinical trial data, not empirical data. If none of the scenarios are accurate approximations to the true dose-efficacy of a vaccine, then the findings and recommendations of this work may not be relevant. We accounted for this weakness by investigating many qualitatively different scenarios.

Secondly, we included only binary measures of efficacy/toxicity. In practice, there may be non-binary outcomes of interest, for example CD4+ T Cell percentage or antibody titres. The CCBP model used for the CoBe DOA can only be used for binary responses, and further work would be needed to investigate similar DOAs for non-binary responses. Vaccine dose-optimisation based on non-binary responses would require more complicated utility functions and models, but this was beyond the scope of this work.

Thirdly we ignored some features that are commonly used in adaptive trial design and may be practically desirable. For example, stopping rules, which are criteria that allow for dose-finding trials to end early if there is sufficient evidence to suggest that one dose is optimal [59]. Additionally, we ignored escalation/de-escalation criteria, which are criteria that limit trial doses to a sub-range of the dosing domain until there is sufficient evidence to support escalating to larger and potentially more toxic doses [30,60]. Both of these trial design features would have added complexity to the implementation of the DOAs. Given that all evaluated DOAs did not include these features, we believe that our results were not biased by this weakness. Further work may need to be conducted to evaluate the effects of including stopping rules or escalation/de-escalation criteria. Similarly, to reduce the scope and complexity of this work, we also only compared the CoBe DOA against three other DOAs. Further work could conduct a comparison with other DOAs, for example rule-based designs [61], or the EffTox [34,35] or Bayesian Optimal Design algorithm [24].

We only investigated one parametric model each for single-administration dose-efficacy, single-administration dose-toxicity, prime/boost administration dose-efficacy, prime/boost administration dose-toxicity, and prime/boost/second-boost dose-efficacy. It is possible that the results for the parametric DOA for some scenarios would have been different if we had chosen different parametric models. For example, for scenario 6 of objective 1, the parametric DOA had a low mean true efficacy relative to the other DOAs after 300 trial participants. It is possible that this was due to that parametric model being misspecified for that scenario, and that a parametric DOA which used a parametric model of undulating dose–response would have been more effective in that scenario. Whilst this may have impacted our results, it also highlights why we believe non-parametric models may have potential for use in vaccine dose-optimisation, as these may have reduced risk of choosing a model that may negatively impact selection of optimal vaccine dose compared to parametric models.

To reduce complexity, we did not evaluate in the main body of this work the effects of changes in cohort size, type of CCPB kernel, the CCBP kernel length hyperparameter values, the ‘temperature’ used for the softmax selection for the Parametric DOA, and the number of doses used for the Uniform Naive and Adaptive Naive DOAs. We do however provide an evaluation of the effects of changing in the Appendix A. Additionally, whilst we showed that it is likely that mathematical modelling and adaptive trial design may lead to selection of more optimal vaccine doses and improve benefit to trial participants, these methods may increase the complexity and duration of conducting vaccine dose-ranging trials. Vaccine developers may need to consider whether this complexity is justified by the potential benefits of more ethical trials and improved clinical vaccine doses.

Comparing our work to previous findings, other DOAs that used non-parametric models were found to be as effective as using parametric modelling based DOAs for the selection of optimal dose [24,25], which is consistent with our findings. The use of mathematical modelling methods and adaptive design has previously been found to lead to more ethical clinical trials [17,62]. This is consistent with our finding that the Uniform Naive DOA, which represents a non-modelling approach, had the lowest cumulative sums of efficacy/utility across most scenarios. In model-based drug development, using beta distributions for dose-ranging adaptive trial design investigating combination oncological drugs has previously been found to be effective through simulation studies [63]. That work however considered only a small number of dosing groups, did not use similarity kernels which were fundamental to the CoBe DOA presented in this work, and chose to use a Jeffreys’ prior rather than a uniform prior for their Beta distributions, and so the methodologies are qualitatively different. This work aligns well with other work in the field of Immunostimulation/Immunodynamic (IS/ID) modelling, which has proposed increased adoption of mathematical modelling for the purposes of optimal vaccine dosing.

There are several future possible areas for research. Firstly, while this work shows a theoretical validation of the CoBe DOA, clearly empirical validation of these methods is required. Secondly, further development and extension of the CoBe DOA would be beneficial to allow for uptake of these methodologies into clinical use. A quantitative method for selecting length hyperparameters for the CCBP model would be beneficial, and is discussed in [28,29]. Whilst inclusion of expert knowledge into CCBP models was evaluated in objective 4, a validated method of extrapolating from animal vaccine dose-efficacy/dose-toxicity data to inform priors for these models would be highly beneficial considering that preclinical data are often used in the design of human dose-ranging trials. This may be important given the findings of objective 4, as for all scenarios the incorrect priors were detrimental to both selection of optimal dose and maximising benefit to trial participants. Further, the evaluation of CoBe DOA using more complicated utility functions of multiple immunological/toxicological responses is needed and may be more informative. Developing computational software such as an R or Python package may also be beneficial for allowing practical application of the CoBe DOA.

There is also potential for future research into other non-parametric models for the purpose of vaccine dose optimisation. Extensions of the CCBP or other non-parametric models should be investigated for modelling continuous or ordinal dose–response data. Additionally, we considered a homogenous trial population in this work. Developing methods of accounting for heterogeneity in the clinical trial population, either through trial participant randomisation or through augments to the models, could be important to ensuring maximal vaccine benefit.

## 5. Conclusions

Selection of optimal vaccine dose is an important but complicated endeavour. In this work we evaluated a novel approach for selecting optimal vaccine dose using the non-parametric CCBP model and adaptive trial design. Using mathematical models and/or adaptive design may lead to more effective and ethical vaccine dose-finding clinical trials, even if the shape of the dose-efficacy curve is unknown. These methods may also maximise benefit to vaccine clinical trial participants. This is the first novel investigation of modelling-based vaccine dose-optimisation approaches when compared to non-modelling vaccine dose-optimisation approaches. If developed further and implemented into vaccine clinical trials, mathematical modelling could accelerate vaccine development and save lives.

## Figures and Tables

**Figure 1 vaccines-10-01838-f001:**
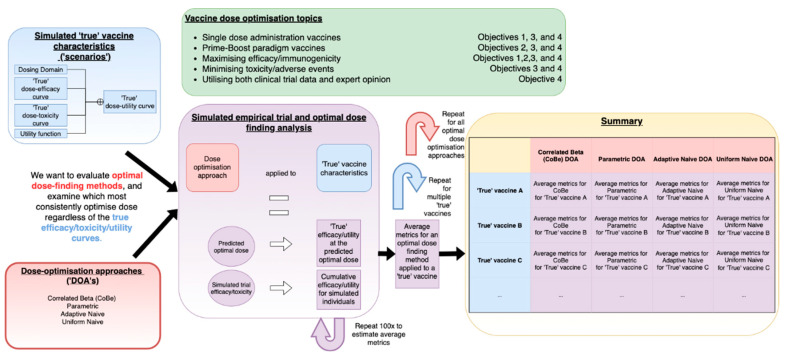
A visual depiction of the process of conducting simulation studies used in this work to evaluate the Correlated Beta (CoBe) and other potential approaches of vaccine dose optimisation (red). These were tested by simulating clinical trials (purple) based on ‘scenarios’ (blue). Repeated simulation of clinical trials was conducted for different dose-optimisation approach/scenario pairs, and metrics related to how effectively optimal dose was located were calculated. These were tabulated and compared to assess these approaches. These analyses were used with considerations of several open topics in vaccine dose-optimisation (green).

**Figure 2 vaccines-10-01838-f002:**
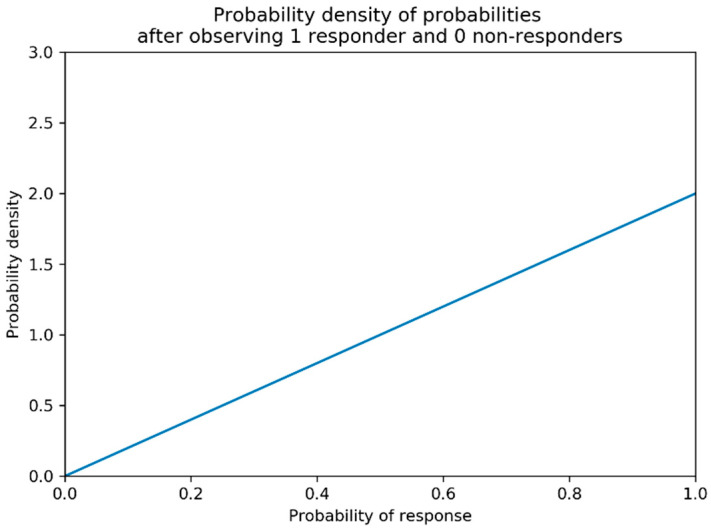
Probability density of probabilities of response after observing one responder and zero non-responders with no prior knowledge regarding probability of response. The higher the probability density is for a given probability of response, the more likely that it is the true probability of response given the data. The above is formally a Beta (2,1) distribution.

**Figure 3 vaccines-10-01838-f003:**
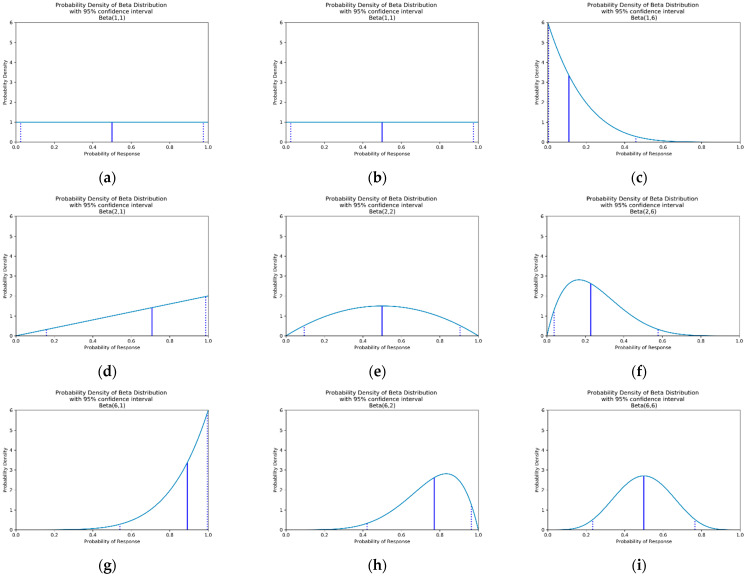
Probability density for beta distributions $Beta\left(\alpha ,\beta \right)$ with differing values of α and β. The vertical blue line represents the median probability of response (efficacy or toxicity), and the dashed vertical lines represent the 95% confidence interval. If using an uninformative prior (**a**), these represent the probability density after observing: (**a**) 0 responders and 0 non-responders, (**b**) 0 responders and 1 non-responders, (**c**) 0 responders and 5 non-responders, (**d**) 1 responders and 0 non-responders, (**e**) 1 responder and 1 non-responder, (**f**) 1 responder and 5 non-responders, (**g**) 5 responders and 0 non-responders, (**h**) 5 responders and 1 non-responder and (**i**) 5 responders and 5 non-responders.

**Figure 4 vaccines-10-01838-f004:**
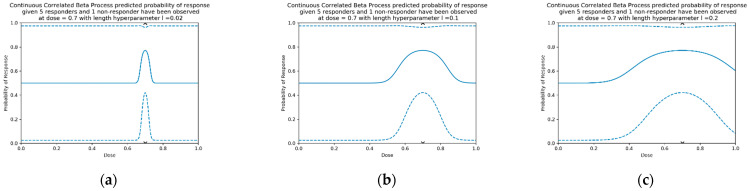
An example of three different CCBP models with squared exponential kernels for different length hyperparameters, using the flat prior for all doses, and 6 observed responses at dose = 0.7 (5 responders, 1 non-responder, as per Figure 1h). Length parameters are (**a**) 0.02, (**b**) 0.1 and (**c**) 0.2. The solid line is the median predicted probability, and the dashed lines show the 95% confidence interval.

**Figure 5 vaccines-10-01838-f005:**
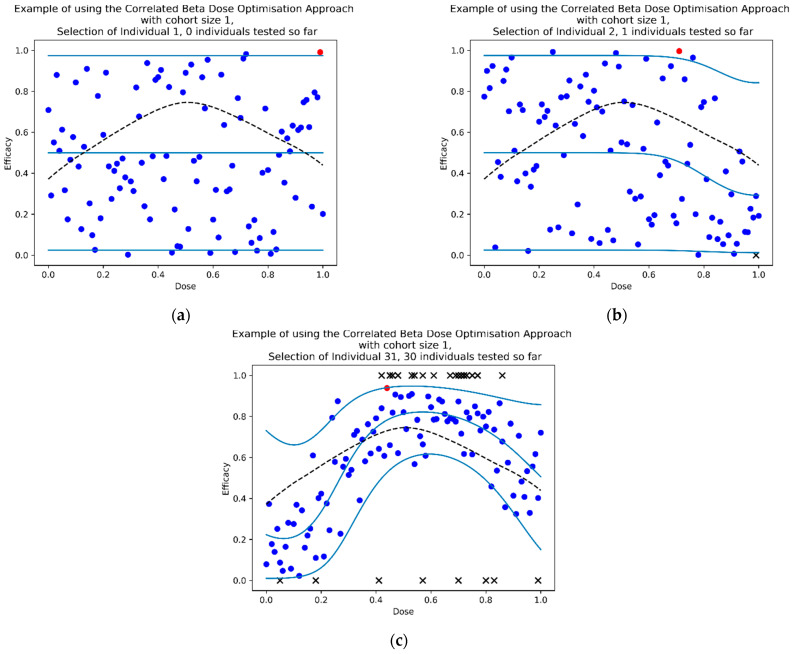
Three timepoints of an example dose-finding study using the CoBe DOA for selection of optimal dose (sampling cohort size c = 1) with the ‘Efficacy Maximisation’ utility function. At each time point, the dotted black line represents the true underlying dose-efficacy curve, and the black crosses represent observed data by that time point. The light blue lines represent the median and 95% confidence intervals (CI) for the predictions of the CCBP model, the blue dots represent efficacy prediction samples for each dose, the red dot represents the maximum of such samples which corresponds to the dose that would be selected to be tested next. At selection of individual 1 (**a**) efficacy samples varied uniformly between 0 and 1. After the first individual received a high dose and efficacy was not observed, (**b**) the median, 95%CI and samples for similar doses were lower when selecting the dose for individual 2. After the 30th individual (**c**), most samples had been selected near the true optimal and the model approximated the true curve (particularly near the predicted optimal dose).

**Figure 6 vaccines-10-01838-f006:**
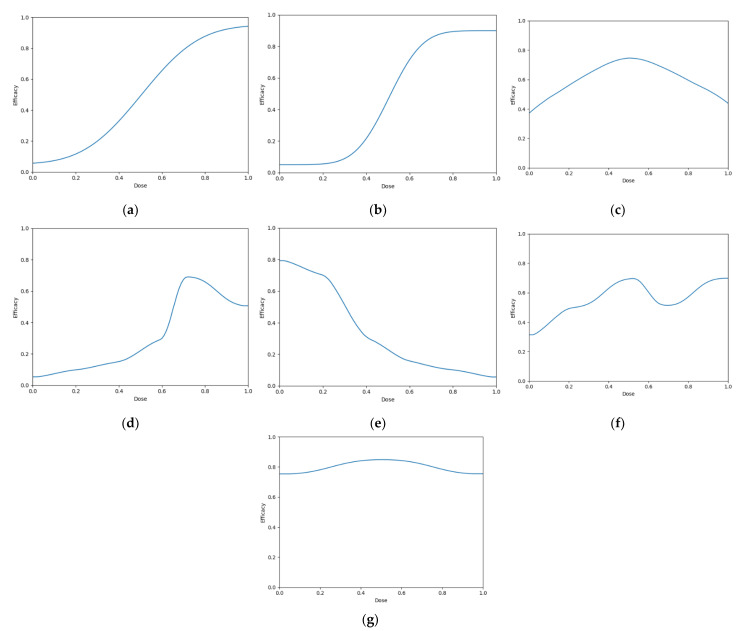
Dose-efficacy plots for the seven objective 1 scenarios. These were (**a**) scenario 1, (**b**) scenario 2, (**c**) scenario 3, (**d**) scenario 4, (**e**) scenario 5, (**f**) scenario 6, and (**g**) scenario 7. Purple represents areas of higher efficacy. The blue line represents the probability of efficacious response for each dose.

**Figure 7 vaccines-10-01838-f007:**
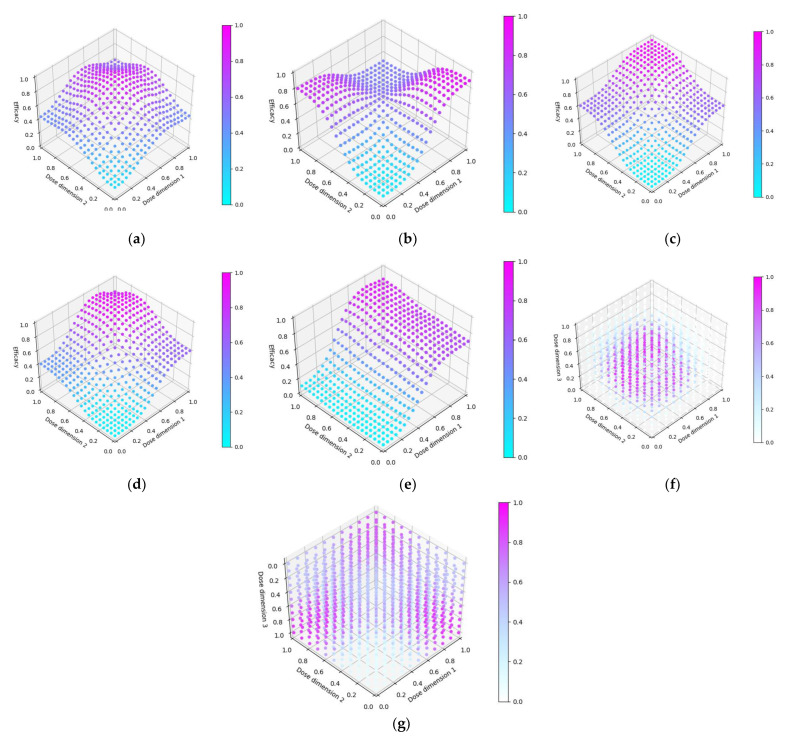
Dose-efficacy plots for the seven objective 2 scenarios. These were (**a**) scenario 1, (**b**) scenario 2, (**c**) scenario 3, (**d**) scenario 4, (**e**) scenario 5, (**f**) scenario 6, and (**g**) scenario 7. Purple represents areas of higher efficacy. Note the *z*-axis for (**g**) is inverted to better show the 3-dimensional dose-efficacy relationship.

**Figure 8 vaccines-10-01838-f008:**
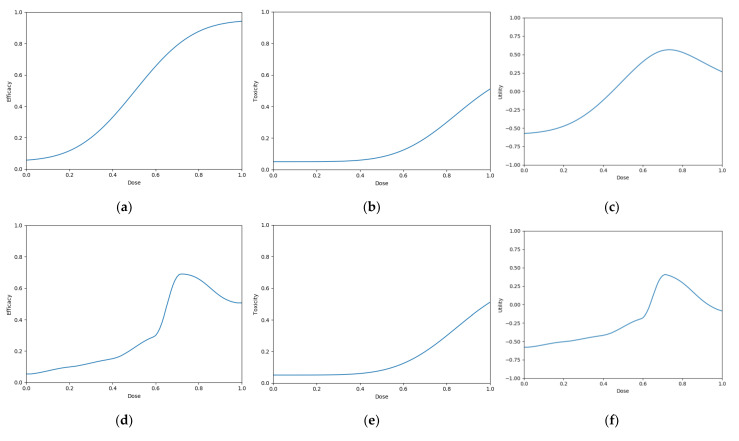
Dose-efficacy/toxicity/utility plots for the six objective 3 scenarios. These were (**a**–**c**) scenario 1, (**d**–**f**) scenario 2, (**g**–**i**) scenario 3, (**j**–**l**) scenario 4, (**m**–**o**) scenario 5, and (**p**–**r**) scenario 6. Left/middle/right plots were, respectively, for efficacy/toxicity/utility. (**a**–**l**) The blue line represents the probability of efficacious/toxic response and utility for each dose. (**m**–**o**) Purple represents areas of higher efficacy/toxicity/utility.

**Figure 9 vaccines-10-01838-f009:**
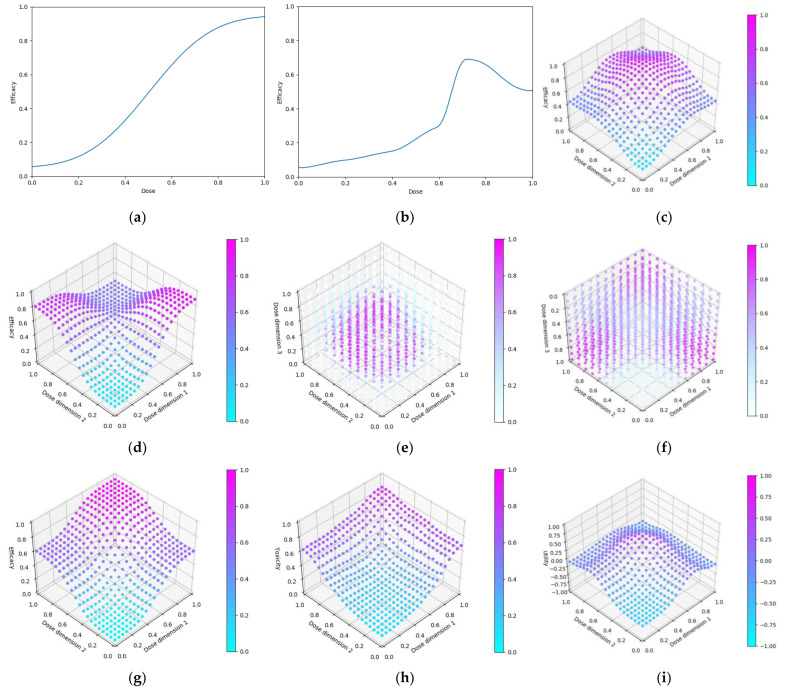
Dose efficacy and efficacy/toxicity/utility plots for the seven objective 4 scenarios. Dose-efficacy plots were (**a**) scenario 1, (**b**) scenario 2, (**c**) scenario 3, (**d**) scenario 4, (**e**) scenario 5, (**f**) scenario 6. For (**g**–**i**) scenario 7, these were, respectively, for efficacy/toxicity/utility. (**a**,**b**) The blue line represents the probability of efficacious/toxic response and utility for each dose. (**c**–**i**) Purple represents areas of higher efficacy/toxicity/utility.

**Figure 10 vaccines-10-01838-f010:**
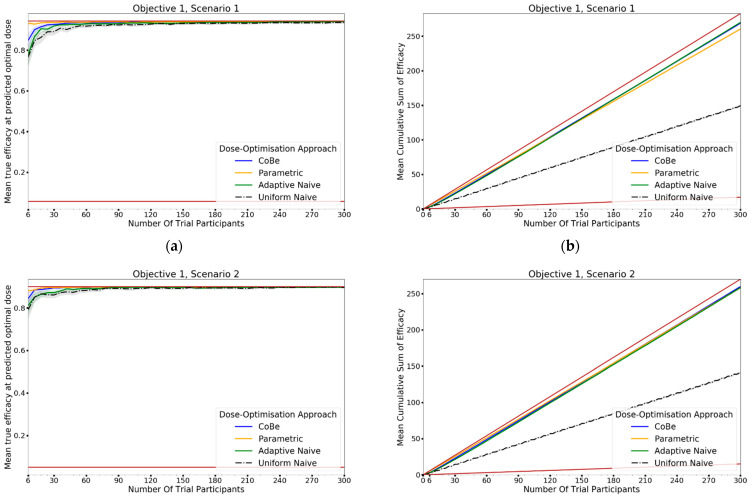
Mean true efficacy at the predicted optimal dose (left) and mean cumulative sum of efficacy (right) against trial size for all seven objective 1 scenarios (Scenario 1 (**a**,**b**), Scenario 2 (**c**,**d**), Scenario 3 (**e**,**f**), Scenario 4 (**g**,**h**), Scenario 5 (**i**,**j**), Scenario 6 (**k**,**l**), Scenario 7 (**m**,**n**)). These are the mean values and 95%CI values across 100 simulations. For the true efficacy plots (left), the brown lines show the minimum and maximum possible efficacy that could be achieved in that scenario. For the cumulative efficacy plots (right) the brown lines represent the maximum and minimum cumulative efficacy sum that could be expected for that scenario. For example, if the true maximum efficacy for a scenario was 90%, no DOA could locate a dose with a true efficacy >90%, and no DOA could achieve a mean cumulative efficacy > 270 (=90% × 300) after testing 300 trial participants. A mean true efficacy/mean cumulative sum of efficacy curve being closer to the upper brown line reflects that DOA being the more effective for locating a maximally efficacious dose/maximising benefit to trial participants.

**Figure 11 vaccines-10-01838-f011:**
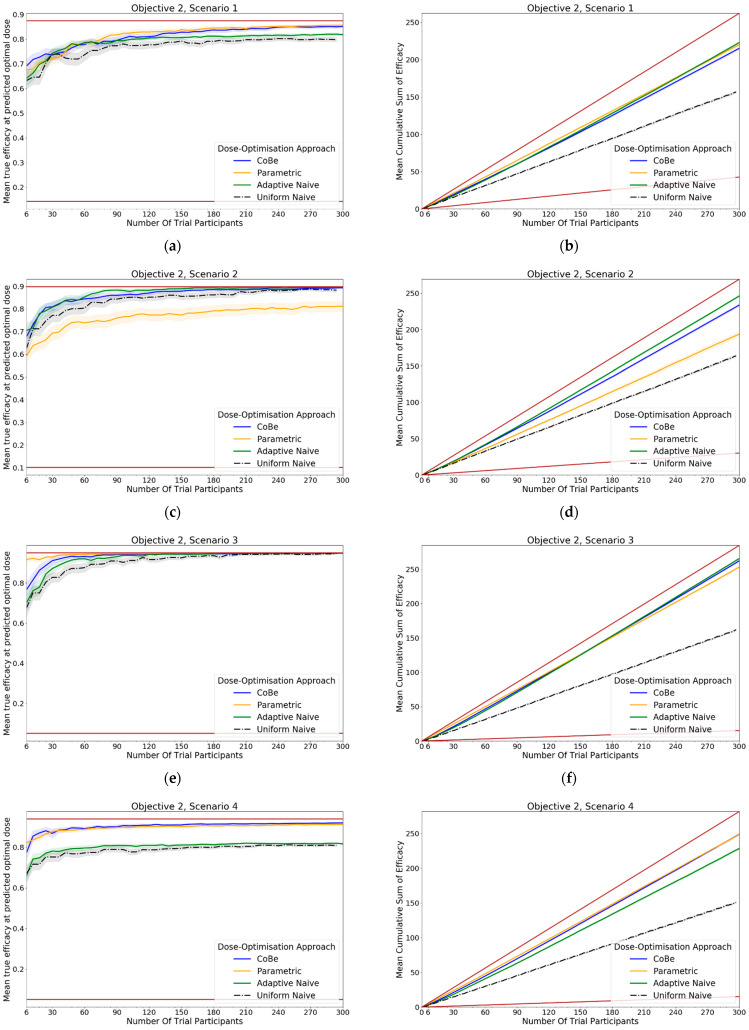
Mean true efficacy at the predicted optimal dose (left) and mean cumulative sum of efficacy (right) against trial size for all seven objective 2 scenarios (Scenario 1 (**a**,**b**), Scenario 2 (**c**,**d**), Scenario 3 (**e**,**f**), Scenario 4 (**g**,**h**), Scenario 5 (**i**,**j**), Scenario 6 (**k**,**l**), Scenario 7 (**m**,**n**)). These are the mean values and 95%CI values across 100 simulations. For the true efficacy plots (left), the brown lines show the minimum and maximum possible efficacy that could be achieved in that scenario. For the cumulative efficacy plots (right) the brown lines represent the maximum and minimum cumulative efficacy sum that could be expected for that scenario.

**Figure 12 vaccines-10-01838-f012:**
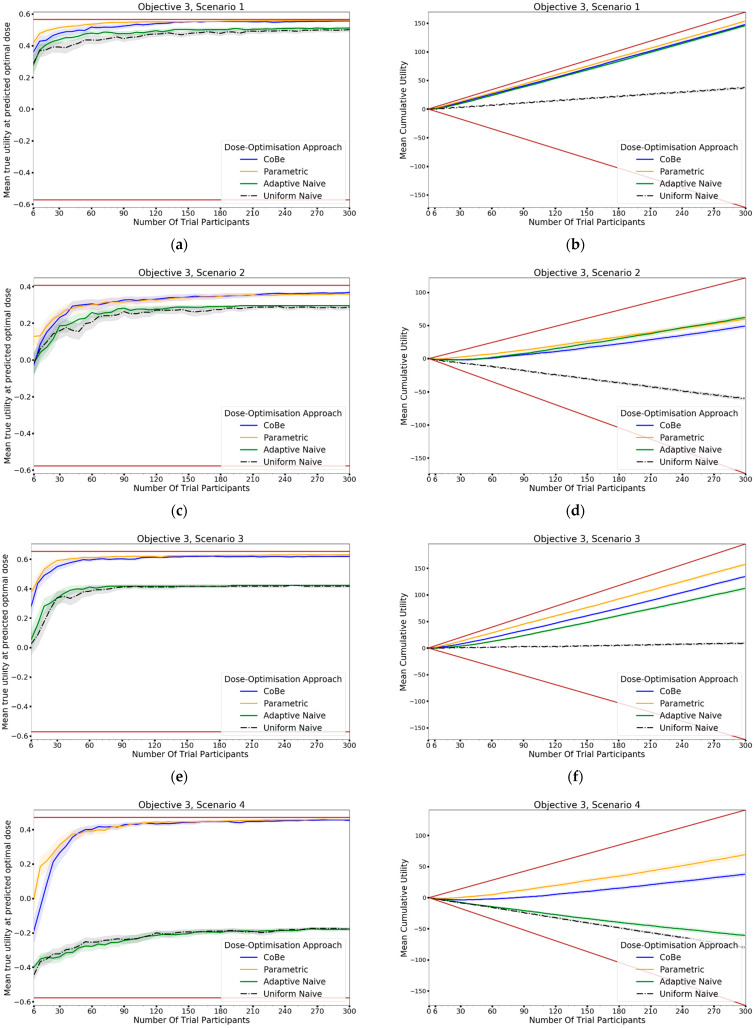
Mean true utility at the predicted optimal dose (left) and mean cumulative sum of utility (right) against trial size for all six objective 3 scenarios (Scenario 1 (**a**,**b**), Scenario 2 (**c**,**d**), Scenario 3 (**e**,**f**), Scenario 4 (**g**,**h**), Scenario 5 (**i**,**j**), Scenario 6 (**k**,**l**)). These are the mean values and 95%CI values across 100 simulations. For the true utility at the predicted optimal dose plots (left), the brown lines show the minimum and maximum possible utility that could be achieved in that scenario. For the cumulative efficacy plots (right) the brown lines represent the maximum and minimum cumulative utility sum that could be expected for that scenario.

**Figure 13 vaccines-10-01838-f013:**
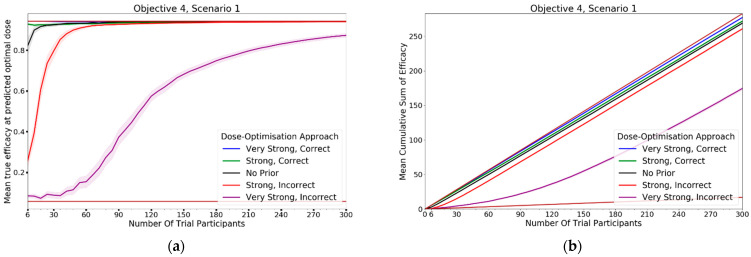
Mean true efficacy at the predicted optimal dose (S1–6, left), mean cumulative sum of efficacy (S1–6 6, right), mean true utility at the predicted optimal dose (S7, left), and mean cumulative sum of utility (S7, right) against trial size for all seven objective 4 scenarios (Scenario 1 (**a**,**b**), Scenario 2 (**c**,**d**), Scenario 3 (**e**,**f**), Scenario 4 (**g**,**h**), Scenario 5 (**i**,**j**), Scenario 6 (**k**,**l**), Scenario 7 (**m**,**n**)). These are the mean values and 95%CI values across 100 simulations. For the true efficacy/utility at the predicted optimal dose plots (left), the brown lines show the minimum and maximum possible efficacy/utility that could be achieved in that scenario. For the cumulative efficacy/utility plots (right) the brown lines represent the maximum and minimum cumulative efficacy/utility sum that could be expected for that scenario.

## Data Availability

Data and code for this work are available through https://github.com/ISIDLSHTM/CoBeDOA_Data (accessed on 24 August 2022) and https://github.com/ISIDLSHTM/CoBeDOA (accessed on 24 August 2022).

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
