# Peer review of "The Correlated Beta Dose Optimisation Approach: Optimal Vaccine Dosing Using Mathematical Modelling and Adaptive Trial Design"

_vaccines, 2022, doi:10.3390/vaccines10111838_

Round 1

Reviewer 1 Report

No comment

Author Response

We thank the reviewer for their time and expertise in reviewing this work.

Reviewer 2 Report

Thank you for an interesting and well-written manuscript presenting a novel adaptive approach to vaccine dose finding and investigating this approach with a very well-described simulation study. I especially appreciated the detailed sensitivity analyses in the appendix. I only have some minor points to consider for improving the clarity of the manuscript:

Title: There is an unexpected "-" in "Vac-Cine"

Formula 7 and 8: It is unclear what "completely similar"/"completely dissimilar" mean in this context.

Formula 10: The K(d_j,d_j) part of the formula seems superfluous, as self-similarity only is a statement about one dose (d_i).

Line 260: It is unclear at this point, why l=0.2 was chosen. This is very well-described in the supplementary, so maybe a reference to the supplementary could be added here.

Formula 16 and 17: The minus between the quotients is hard to read, maybe some extra space could be added. (I read it as one long quotient first, as it looked like a single long horizontal line.)

Line 279-281: Again it is unclear how the l-values were chosen, a reference to the discussion in the supplementary would be helpfull.

Line 215: Why is the median used (and not e.g. the mode)?

Line 216: Maye it here should be "probability of efficiacy and toxicity"?

Line 234: Shouldnøt it be "are" instead of "is" here?

Algorithm 1: It is unclear here, why step 7 is done each iteration and not only once at the end. This makes sense later on in the manuscript, as you report the efficiacy during the trials, but maybe it could be added as an explanation here to avoid confusing the reader?

Formula 24: There is an unexpected ' after the z.

Line 360: Shouldn't the formnula also be given for the setting with two boosters? (It is straightforward, so maybe just a statement, that it is similar would work as well.)

Line 527 anf following: The numbering of these points is confusing (similar for later lists: line 582, line 649)

Line 629: What is "top"?

Figure 9: It seems like the y-axis of (e) and (f) should be "Dose dimension 3" and not "Efficacy".

Line 693 and 699: These two paragraphs seem to conflict. The first states that CoBe was similar or better for all scenarios, but the next states that it was sometimes worse. (A similar problem arises in line 791 and 784).

Line 771-780: This section repeats much from the section before. (Similar repititions occur in line 795-799, line 856-872, and line 920-932)

Line 853: Here the statement "Cumulative sum of efficacy" seems like an unplanned repitition of the following heading.

Line 900: Aren't there five and not only four DOAs?

Line 991: I think it should be "few" and not "small" in this case.

Supplementary S5: You write "I" instead of "we" (used otherwise) on the bottom of the page.

Supplementary S5: A "for" seems to much just before the list of inputs.

Supplementary S6: It porbaby should be "shows" and not "show" above the first table.

Supplementary S6: In the "Objective 1, Scenario 2 Efficacy" table you state -0.0. This of course equals 0.0, but the minus is surprising, is there a typo?

Supplementary S6: In the "Objective 2, Scenario 1 Efficacy" table and many of the following some of the decimal numbers are not starting with a 0, which makes it harder to read (especially as it is inconsistent), e.g. ".9" instead of "0.9".

Supplementary S7.1: Is a symbol (maybe =) missing in the K(di,di+0.15)0.5 formula?

Supplementary S7.2: At one point towards the end you write 3-D with a minuscle "d".

Supplementary S7.2: In the second to last paragraph you twice write "I" instead of "we".

Supplementary S7.5: Shouldn't it be "Uniform Naive DOAs" in the statement "CoBe DOAs with six different values for b"?

ISIDLSHTM/CoBeDOA_Data repository: A link to the main repo in the description would be helpfull. (Similar as the main repo has a linl to the data repo.) Furthermore, "separately" has a typo in the description. (Note that I didn't run the code, just briefly looked into it.)

Author Response

Thank you for an interesting and well-written manuscript presenting a novel adaptive approach to vaccine dose finding and investigating this approach with a very well-described simulation study. I especially appreciated the detailed sensitivity analyses in the appendix. I only have some minor points to consider for improving the clarity of the manuscript:

We thank the reviewer for their time and expertise in reviewing this work. We are glad that the sensitivity analyses were appreciated, and appreciate the detailed comments which we believe will have improved the work. 

Title: There is an unexpected "-" in "Vac-Cine"

This has been corrected.

Formula 7 and 8: It is unclear what "completely similar"/"completely dissimilar" mean in this context.

We have added the following text to this section to further define these theoretical concepts:

‘where ‘complete similarity’  would imply that clinicians/modellers believe that observing response/non-response for dose d_i  is equivalent to observing response/non-response for dose d_j  for the purposes of predicting response probability for dose d_j. Likewise, ‘completely dissimilar’  would imply that clinicians/modellers do not believe that observing a response/non-response for dose d_i provides any information regarding response probability for dose d_j.'

Formula 10: The K(d_j,d_j) part of the formula seems superfluous, as self-similarity only is a statement about one dose (d_i).

This has been corrected. The K(d_j,d_j) was superfluous, and we agree that the formula may be simpler to parse with only d_i

Line 260: It is unclear at this point, why l=0.2 was chosen. This is very well-described in the supplementary, so maybe a reference to the supplementary could be added here.

Reference to S2 and S7.1 have been added.

Formula 16 and 17: The minus between the quotients is hard to read, maybe some extra space could be added. (I read it as one long quotient first, as it looked like a single long horizontal line.)

Additional space between the quotients has been added. We thank the reviewer, and agree that this space makes the formula easier to read.

Line 279-281: Again it is unclear how the l-values were chosen, a reference to the discussion in the supplementary would be helpfull.

This has been added

Line 215: Why is the median used (and not e.g. the mode)?

We thank the reviewer for this query, which we do believe is an interesting topic and was a consideration that we had made previously. This figure was presented to aid readers in feeling comfortable with beta distributions, and so we felt that the choice of showing medians and 95% intervals reflects that these are Bayesian beliefs regarding probability. With the median, we highlight more explicitly the value for which we believe there is a 50% probability that the true probability of response is greater than, and 50% probability that the true probability of response is greater than. This is similar to the confidence bounds, which show the interval that we are 95% certain that the true probability of response is between given our prior and observable data. 

Whilst the mode may have been reasonable (and is indeed the maximum a-posteriori estimate), there are features of the mode which could be confusing for this figure. In particular, for (b,c,d,e) the modal value would equal 0 or 1 and be outside of the 95% interval, which may be confusing to some readers. We felt this would be contradictory to the aim of this explanatory figure. 

We note further that, as beta distributions can be skewed (f,h), the median may better reflect the distribution than the mean/mode. Whilst beta distributions have the unfortunate property that an exact closed-form solution for the median is not in general available, we did believe that this was the most parsimonious measure of the distribution to highlight.

We also note that as alpha and beta increase, the mean, median and mode tend towards the same value.  

We have not made any changes to the work in response to this comment, but hope that the reviewer is comfortable with our reasoning.

Line 216: Maye it here should be "probability of efficiacy and toxicity"?

This has been added to specify what was meant.

Line 234: Shouldnøt it be "are" instead of "is" here?

We apologise, the line numbers may have been altered between versions and so we could not find an ‘is’ that we believed would be better replaced with ‘are’ in that paragraph. We replaced an ‘is’ with ‘exists’ in the following sentence, and hope that this was the improvement desired. 

‘Typically, if there exists no prior knowledge for which response probabilities are most reasonable, it is best to use an uninformative prior. 

Algorithm 1: It is unclear here, why step 7 is done each iteration and not only once at the end. This makes sense later on in the manuscript, as you report the efficiacy during the trials, but maybe it could be added as an explanation here to avoid confusing the reader?

We thank the reviewer for noting this and for highlighting why we chose to conduct step 7 at each cohort. We agree with the reviewer, and have added the following text to section 2.2.5. 

‘We note that, in practical application, clinicians/modellers may choose to skip step 7 of this algorithm until the final sampling cohort is completed. However, in this work this step was conducted after each cohort to investigate the effect of increasing trial size on improvement in vaccine dose selection.’

Formula 24: There is an unexpected ' after the z.

This has been corrected

Line 360: Shouldn't the formnula also be given for the setting with two boosters? (It is straightforward, so maybe just a statement, that it is similar would work as well.)

In this work scenarios investigating prime/boost/second-boost vaccines were only conducted for ‘Efficacy Maximisation’ and so we did not model two-booster toxicity for any scenarios. As noted by the reviewer, defining such a model would be straightforward. We believe that, as such a model was not used, it is preferable to not give the equation for one.

Line 527 anf following: The numbering of these points is confusing (similar for later lists: line 582, line 649)

We have corrected this numbering issue.

Line 629: What is "top"? 

This should be ‘to’ rather than ‘top’. This has been corrected, and we thank the reviewer for noticing this error.

Figure 9: It seems like the y-axis of (e) and (f) should be "Dose dimension 3" and not "Efficacy".

This has been corrected. 

Line 693 and 699: These two paragraphs seem to conflict. The first states that CoBe was similar or better for all scenarios, but the next states that it was sometimes worse. (A similar problem arises in line 791 and 784).

We thank the reviewer. This has been corrected in both instances, with ‘all trial sizes’ replaced with ‘most trial sizes’ to better reflect the results. 

Line 771-780: This section repeats much from the section before. (Similar repititions occur in line 795-799, line 856-872, and line 920-932)

We thank the reviewer for this comment. We agree that this is largely a repetition of ideas, but given the complexity and length of the work. Early drafts of the work did not include this repetition, and we found that this led to confusion from test readers. We believe that this repetition may be useful in ensuring that for each section of the results the reader is best prepared to understand the results. Therefore this repetition was intentional, and we have made no amendments in response to this comment.

Line 853: Here the statement "Cumulative sum of efficacy" seems like an unplanned repitition of the following heading.

This has been corrected.

Line 900: Aren't there five and not only four DOAs?

This has been corrected.

Line 991: I think it should be "few" and not "small" in this case.

This has been corrected.

Supplementary S5: You write "I" instead of "we" (used otherwise) on the bottom of the page.

This has been corrected.

Supplementary S5: A "for" seems to much just before the list of inputs.

This has been corrected.

Supplementary S6: It porbaby should be "shows" and not "show" above the first table.

This has been corrected

Supplementary S6: In the "Objective 1, Scenario 2 Efficacy" table you state -0.0. This of course equals 0.0, but the minus is surprising, is there a typo?

This has been corrected

Supplementary S6: In the "Objective 2, Scenario 1 Efficacy" table and many of the following some of the decimal numbers are not starting with a 0, which makes it harder to read (especially as it is inconsistent), e.g. ".9" instead of "0.9".

This section has been generally improved following this comment, including 0s to start decimals and improving consistency in spacing.

Supplementary S7.1: Is a symbol (maybe =) missing in the K(di,di+0.15)0.5 formula?

The ‘=’ was missing, we thank the reviewer for noticing this error.

Supplementary S7.2: At one point towards the end you write 3-D with a minuscle "d".

This has been corrected.

Supplementary S7.2: In the second to last paragraph you twice write "I" instead of "we".

This has been corrected.

Supplementary S7.5: Shouldn't it be "Uniform Naive DOAs" in the statement "CoBe DOAs with six different values for b"?

This has been corrected.

ISIDLSHTM/CoBeDOA_Data repository: A link to the main repo in the description would be helpfull. (Similar as the main repo has a linl to the data repo.) Furthermore, "separately" has a typo in the description. (Note that I didn't run the code, just briefly looked into it.)

A link has now been added, and the spelling corrected.

We again sincerely thank the reviewer for their detailed and insightful comments, which we believe reflect a thorough reading of this work and have improved the quality of the manuscript.

Reviewer 3 Report

In this paper, the authors aimed to use simulation of dose-finding clinical trials to assess the use of the ‘Correlated Beta dose optimisation approach’ in selecting optimal vaccine dose. The paper can be accepted after some minor revisions.

1. Some acronyms should be defined.

2. In introduction section, give more motivation and novelty of your study.

3. Some figures are not clear.

4. Make a comparison between your results and other findings in the              literature.

5. There are some typos. The authors should carefully read the manuscript.

Author Response

We thank the reviewer for their time and expertise in reviewing this work. We appreciate the comments which we believe will have improved the work.

1. Some acronyms should be defined.

We have been through the document to ensure this is true, and have added definition where it was not. We include a list of acronyms that we believe were used in this manuscript. 

    • CoBe - First used and defined on line 73.
    • DOA - First used and defined on line 87.
    • CCBP - First used and defined on line 65.
    • LHS - First used and defined on line 730.
    • RHS - First used and defined on line 760.
    • IS/ID - First used and defined on line 1094.
    • CI - First used and defined on line 349.

2. In introduction section, give more motivation and novelty of your study.

We have added the following text to the introduction to highlight motivation and novelty.

‘We hoped that this novel modelling approach could have potential for practical application over a number of vaccine use cases, and that the highlighted model could provide interpretable quantitative insight for vaccine developers’ 

3. Some figures are not clear.

We have increased the DPI for figures 1-5, which were lower quality. 

4. Make a comparison between your results and other findings in the literature.

We have added reference to additional work in a similar field. 

‘In model-based drug development, using beta distributions for dose-ranging adaptive trial design investigating combination oncological drugs has previously been found to be effective through simulation studies [63]. That work however considered only a small number of dosing groups, did not use similarity kernels which were fundamental to the CoBe DOA presented in this work, and chose to use a Jeffreys’ prior rather than a uniform prior for their Beta distributions, and so the methodologies are qualitatively different.’

5. There are some typos. The authors should carefully read the manuscript.

The manuscript has been read through and typos/grammar corrected.

 We thank the reviewer for their comments.